# PROCESS-LEVEL TRAJECTORY EVALUATION FOR ENVIRONMENT CONFIGURATION IN SOFTWARE ENGINEERING AGENTS

**Jiayi Kuang**[1,2,*] **Yinghui Li**[2,†] **Xin Zhang**[2,3,*], **Yangning Li**[2,4,*], **Di Yin**[2], **Xing Sun**[2],
**Ying Shen**[1,4,6†], **Philip S. Yu**[5]
[1]Sun Yat-sen University, [2]Tencent Youtu Lab, [3]The Hong Kong Polytechnic University
[4]Peng Cheng Laboratory, [5]University of Illinois Chicago
[6]Guangdong Provincial Key Laboratory of Fire Science and Intelligent Emergency Technology

## ABSTRACT

Large language model-based agents show promise for software engineering, but environment configuration remains a bottleneck due to heavy manual effort and scarce large-scale, high-quality datasets. Existing benchmarks assess only end-to-end build/test success, obscuring where and why agents succeed or fail. We introduce the **En**vironment **Con**figuration **Dia**gnosis Benchmark, `EnConda-Bench`, which provides process-level trajectory assessment of fine-grained agent capabilities during environment setup-**planning**, **perception**-driven error diagnosis, **feedback**-driven repair, and **action** to execute final environment configuration [1]. Our task instances are automatically constructed by injecting realistic README errors and are validated in Docker for scalable, high-quality evaluation. `EnConda-Bench` combines process-level analysis with end-to-end executability to enable capability assessments beyond aggregate success rates. Evaluations across state-of-the-art LLMs and agent frameworks show that while agents can localize errors, they struggle to translate feedback into effective corrections, limiting end-to-end performance. To our knowledge, `EnConda-Bench` is the first framework to provide process-level internal capability assessment for environment configuration, offering actionable insights for improving software engineering agents.

## 1 INTRODUCTION

Large language models (LLMs) have rapidly advanced (Li et al., 2023; Yu et al., 2024; Du et al., 2024; Li et al., 2024c; Huang et al., 2024; Li et al., 2024a; Zhang et al., 2025; Xu et al., 2025; Li et al., 2025e; Kuang et al., 2025b; Team, 2025; Li et al., 2025c;d; Kuang et al., 2025a), spurring exploration of challenging Software Engineering (SWE) tasks with high academic and industrial value (He et al., 2025; Wang et al., 2024; Fan et al., 2023; Wang et al.; Zhang et al., 2024b). SWE offers precise, verifiable evaluation systems, making it a prime domain to study agentic intelligence (Hendrycks et al.; Austin et al., 2021). Numerous code-oriented agents, such as `OpenHands` (Wang et al.) and `Swe-Agent` (Yang et al., 2024), aim to assist with complex project development and maintenance. In SWE benchmarks like SWE-BENCH (Jimenez et al., 2024), agents edit and repair code based on a given issue, then submit a pull request and validate execution. Within this workflow, configuring a runnable execution environment is the most fundamental and critical first step, yet it remains challenging for both human engineers and current LLMs (Eliseeva et al.), requiring substantial manual effort. This burden constrains large-scale, high-quality dataset production, making rigorous evaluation of agents' environment configuration capabilities essential for progress in SWE.

Most existing environment configuration benchmarks rely on end-to-end success (build and test pass) (Milliken et al., 2025; Bouzenia & Pradel, 2025; Eliseeva et al.; Vergopoulos et al., 2025), *yielding only coarse outcomes and obscuring process-level capabilities along the configuration tra-*

---

[*]Work done during the internship at Tencent Youtu Lab.
[†]Correspond to Yinghui Li (lebronyhli@tencent.com) and Ying Shen (sheny76@mail.sysu.edu.cn).
[1]Data and code are available in https://github.com/TencentYoutuResearch/EnConda-Bench.

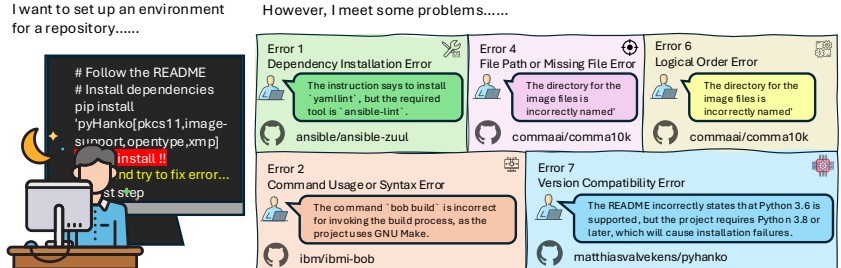

Figure 1: The illustration of common problems in environment configuration. When human engineers configure the environment, they often encounter various errors. They should first identify the step where the error occurred and then fix the problem before proceeding to the next step, until the configuration is complete. Similarly, intelligent agents performing environment configuration should possess good planning, perception, feedback, and action capabilities.

*jectory*. For example, it is difficult to locate the specific stages in environment configuration where errors are likely to occur, or to identify which capabilities the agent lacks to perform more precise and effective configuration. They cannot pinpoint failure stages or missing capabilities, limiting deep insights and research directions. In addition, *data construction is another bottleneck*: high-quality, correctly buildable repositories are scarce; selecting and annotating them demands expert effort. As a result, it is challenging for researchers to obtain large quantities of high-quality data for evaluating agent environment configurations.

To address these challenges, we focus on process-level evaluation along the agent's configuration trajectory. Specifically, we investigate: (1) how agents apply **planning** to devise reasonable configuration steps and strategies given the task requirements; (2) how they use **perception** to accurately localize the causes of errors when failures occur (e.g., version incompatibilities or missing dependencies); (3) how they utilize **feedback** to analyze the errors and try to fix them; and (4) how they translate precise feedbacks into **actions** that correct these errors, complete environment configuration, and ensure that subsequent code runs and passes evaluation. This process-level trajectory evaluation provides deeper and more valuable references for improving agent capabilities in environment configuration and for subsequent related studies.

However, directly extracting the planning and feedback segments from agent trajectories, or evaluating entire long trajectories, is difficult. Inspired by how human engineers configure environments—typically following README steps first, then analyzing the causes of failures and attempting fixes—we consider editing an originally correct README by injecting erroneous commands or confusing steps. As the model configures the environment based on such a README, it must locate and repair these errors. This design enables process-level evaluation along the agent's trajectory and allows us to observe which error types the model more readily repairs and which are harder to detect, providing valuable insights for future agent development.

Motivated by this task schema, we further design an automated data construction framework that scales instance generation and produces agent execution trajectories for training. We (1) select high-quality repositories via strict criteria; (2) employ advanced LLMs to edit key environment READMEs with common error types and annotate categories and suggested fixes; and (3) validate and filter for effective errors via an automated framework to obtain high-quality task instances. We then build an evaluation suite supporting both process-level analysis (error localization, repair) and end-to-end executability, along with an automatic data engineering pipeline that generates task instances and agent trajectories. To our knowledge, we are the first to enable process-level assessment for agents and to propose an automated data framework in this setting. Empirical evaluation on advanced LLMs and agents shows that, while agents exhibit basic error judgment/localization, they struggle to convert feedback into effective corrective actions, limiting end-to-end performance. We present our contributions and several noteworthy findings:

- We propose a trajectory-based `EnConda-Bench` for process-level evaluation of environment configuration in SWE, enabling detailed assessment of the capabilities agents exhibit during environment configuration.

- We introduce an automated data construction pipeline, which reduces manual labor and supplies large-scale training data for agents and LLMs.
- Our evaluation across multiple LLMs/agents finds basic error localization/classification abilities but limited environmental interaction and feedback utilization, often yielding ineffective repairs, providing valuable findings and inspiration for future research.

## 2 RELATED WORK

**Agent Methods** Early agent attempts to automate environment setup relied on specific heuristics that infer dependencies from source code, offering determinism but falling short on system packages, version pinning, and platform heterogeneity (Gruber & Fraser, 2023; Zhang et al., 2024a; Li et al., 2025b; Ye et al., 2025; Yang et al., 2025; Guo et al., 2026; Li et al., 2025a; Lu et al., 2025a;b). Tool-augmented code agents extend LLMs with search, editing, and execution capabilities and show promise (Wang et al., 2024; Wang et al.; Zhang et al., 2024b; Yang et al., 2024; Xia et al., 2024), yet setup remains a fragile bottleneck due to sensitivity to external toolchains and long decision chains. Specialized environment agents try to narrow this gap. INSTALLAMATIC targets Python with curated installation context and exemplar Dockerfiles, judging success via tests (Milliken et al., 2025). EXECUTIONAGENT generalizes to five languages with CI-log ground truth, requiring both Dockerfiles and setup scripts, and evaluating build success and test-result deviations (Bouzenia & Pradel, 2025), but still needs manual inspection and is comparatively slow. Repo2Run employs a dual-environment architecture, performing configurations in an isolated Docker environment while leveraging an external environment for monitoring and assistance, with a rollback mechanism that restores the system to the last known stable state upon command failures (Hu et al., 2025). Overall, the trajectory moves from heuristics to tool-augmented agents to interaction agents, and our approach aims to improve process-level, actionable interactions with the environment.

**Environment Configuration Benchmarks** In early SWE benchmarks, function-level benchmarks (e.g., HumanEval, MBPP, APPS) catalyzed progress but are misaligned with real-world build (Chen et al., 2021; Hendrycks et al.; Li et al., 2022; 2024b; Austin et al., 2021; Jain et al.; Miao et al., 2025; Chen et al., 2025). Repository-level efforts better reflect practice (Liu et al., 2024b; Jain et al., 2024; Jimenez et al., 2024), but they ignore the environment configuration task by providing manually configured Docker files. Environment setup specific benchmarks are attempting to explore this territory. INSTALLAMATICbench curates 40 Python repositories with exemplar Dockerfiles, assessing success via tests (Milliken et al., 2025). EXECUTIONAGENTbench spans five languages with CI-log ground truth and evaluates build/test success and test-result deviations (Bouzenia & Pradel, 2025). For larger-scale data and more languages, recent benchmark EnvBench expands to 994 repositories across Python, Java, and Kotlin projects, while still offering limited visibility into data collection and evaluation strategies (Eliseeva et al.). Thus, automated construction further scales evaluation. SETUPAGENT further automates extraction of installation and testing procedures, supports historical states, and collects test-level results, accelerating data generation, though its evaluation remains largely end-to-end (Vergopoulos et al., 2025). Nonetheless, most benchmarks still reduce evaluation to end-to-end executability, obscuring where and why setup fails. In contrast, our work provides process-level trajectory evaluation with an automatic data construction framework for robust evaluation of agents.

## 3 ENCONDA-BENCH

### 3.1 TASK DEFINITION AND WORKFLOW

As illustrated in Figure 2, EnConda-Bench requires agents to diagnose and repair environment configuration errors. Specifically, when an error arises, the agent should (i) identify the step at which the failure occurs, (ii) analyze the precise error type, and (iii) plan an appropriate repair strategy. Building on this, the agent should refine its feedback and corrective actions to ultimately produce an accurate shell script that fully configures the environment. For evaluation, we assess both (a) whether the environment is successfully built and executable, and (b) whether the agent's trajectory demonstrates correct error localization, reasoning, and feedback usage. Concretely, the task design and full pipeline comprise three components: input task instances, agent execution, and evaluation.

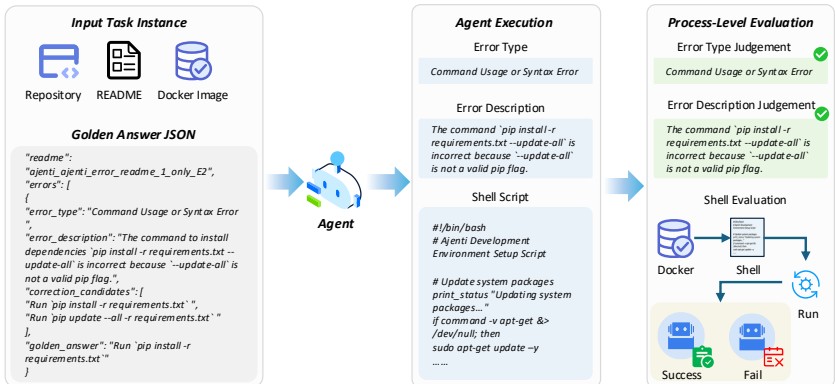

Figure 2: An example of the overall workflow of our process-level environment configuration task.

**Input Task Instances** Each task instance includes: (1) **Repository.** We collect and filter a set of high-quality GitHub repositories to ensure reproducibility and moderate difficulty. To avoid version drift during evaluation, each repository is pinned to a specific *revision commit*. (2) **Dockerfile.** Following EnvBench (Eliseeva et al.), we supply a base Docker image with minimal prerequisites (e.g., Python, Conda). We run the agent to execute the environment configuration inside a Docker container. (3) **README.** Both humans and agents typically begin from the repository's README for environment configuration. Accordingly, each task instance includes the README as the primary guide for the agent's execution. (4) **Labeled golden answer JSON.** For each task instance, we provide a JSON file to support evaluation, including the golden answers of error types, detailed error descriptions, candidate repair command sets, and the final correct command sequence.

**Agent Execution** For each instance, the agent leverages its **planning** abilities to devise a sequence of environment configuration steps, guided by the provided README. Leveraging its **perception** capability, the agent carefully examines the README and repository to identify potential errors. When encountering errors, it employs **feedback**, and analytically reasons them in detail and formulates appropriate repair strategies. Drawing on its **action** skills, the agent implements the proposed fixes and generates a shell script for the environment setup. After execution, we process the trajectory and extract error type judgments, repair commands, and the final shell script.

**Process-Level Evaluation** Given the judgments extracted from the trajectory and the final shell script, we conduct two complementary methods for process-level evaluation. For error diagnosis, we compare the predicted error types, descriptions, and fix suggestions with the gold-standard JSON and compute the corresponding metrics. For executability, we pull the Docker and repository, run the agent's shell script, and check whether it successfully builds the environment and passes the unit test. This evaluation suite yields a process-level assessment of the agent's capabilities for environment configuration, highlighting which capabilities of agents are weaker, which error categories are more easily detected, and which are more challenging.

## 3.2 Data Construction

**Repository Selection** Although GitHub hosts numerous repositories, many do not meet the requirements for reliable environment configuration. If a repository is not reliable (e.g., due to a faulty README or missing dependencies), error annotation becomes labor-intensive and unreliable, and the resulting task may be prohibitively difficult. We therefore retain repositories that satisfy the following criteria that indicate higher quality: at least 10 stars, over 1,000 commits, and more than 10 closed issues. Furthermore, we incorporate repositories from existing benchmarks that have undergone strict human filtering and manual verification of environment setup, using them as the basis for subsequent error synthesis. Details about the repository selection are in Appendix A.1.

**Error Synthesis** After collecting high-quality repositories, we edit the READMEs to synthesize realistic, commonly encountered configuration errors. In fact, our initial plan does not involve syn-

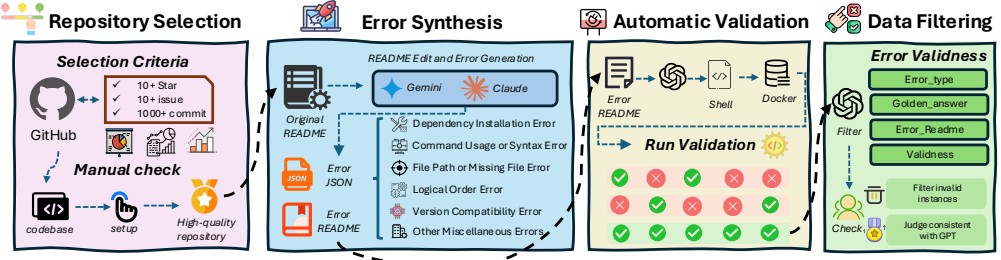

Figure 3: The illustration of our overall pipeline of benchmark construction.

thesizing errors. Instead, we consider leveraging existing READMEs by decomposing them into executable steps (to assess whether each step runs correctly) or by annotating intrinsically error-prone steps. However, this approach is highly labor-intensive. Without step or error annotations, just using tools to conduct evaluation of overall agent trajectories would over-rely on the models themselves, making it difficult to extract key steps from long trajectories and to explore specific capabilities. In addition, each repository typically contains only a single README, and high-quality repositories are scarce, which constrains the number of available task instances.

To address these issues, we treat each executable README as ground truth and inject errors. This enables scalable, automated task generation and supports process-level evaluation of planning, perception, feedback, and action during environment configuration. We define six canonical error categories: *Dependency Installation Error*, *Command Usage or Syntax Error*, *File Path or Missing File Error*, *Logical Order Error*, *Version Compatibility Error*, and *Other Miscellaneous Errors* (see Appendix A.2 for detailed definitions and examples). For each README, we prompt claude-4-sonnet and gemini-2.5-pro to introduce two errors and produce a structured JSON with the error type, description, candidate fixes, and ground truth, while instructing minimal edits limited to the necessary lines, avoiding broad rewrites that could compromise README integrity (detailed settings in the Appendix A.3). Taking strictly filtered original READMEs as reference, each case will yield a controlled error label, a concrete description, and a correct fix. From 323 repositories, we produce 1,772 erroneous READMEs, and each README contains exactly two injected errors.

**Automatic Validation** We then automatically validate the effectiveness of injected errors. An injected error is considered *effective* if: (i) following the erroneous README, the environment setup fails, and (ii) after repairing this error, the setup proceeds through the affected step. For each erroneous README, we use gpt-4.1-mini to generate a shell script and execute it inside the provided Docker environment (details in Appendix A.4). If the script succeeds in building the environment and passes the test, the corresponding error is regarded as invalid. We intentionally avoid stronger models at this stage because they may implicitly "auto-fix", resulting in scripts that diverge from the erroneous README and thus undermine verifying the error's effectiveness.

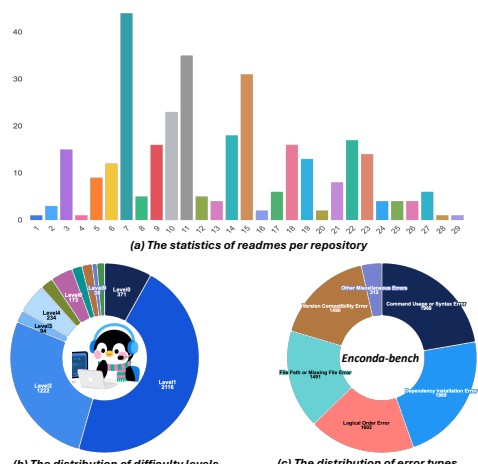

(a) The statistics of readmes per repository

(b) The distribution of difficulty levels

(c) The distribution of error types

Figure 4: Data statistics results.

**LLM-Assisted Filtering and Human Validation**
Automated validation does not guarantee that the injected error is valid, so we conduct a second-pass validation to ensure (a) the error truly impacts configuration, (b) the error is explicitly manifested in the README, and (c) the categorization and proposed fixes are correct. To reduce human effort, we employ gpt-4.1-mini with predefined criteria (see details in Appendix A.5) to assess. The items that fail in this filtering are removed. Human evaluators then review the remaining data under

the same criteria. The agreement between the LLM filter and human judgments reaches 98.5%, supporting the feasibility of this procedure. We ultimately preserve 1,230 valid erroneous READMEs, each containing two errors. To diversify difficulty, we further split and merge errors to construct READMEs containing 1–10+ errors, yielding 4,201 READMEs and 9,471 total errors.

## 3.3 DATASET STATISTICS AND DATA EVALUATION

**Data statistics.** As described in Section 3.2, we complete the benchmark construction. From 323 repositories, we construct 4,201 READMEs, averaging 13 per repository, the distribution shown in Figure 4(a). We further stratify difficulty by the number of injected errors per README, defining levels 1–10, shown in Figure 4(b). Most READMEs fall into level 1 or level 2, which aligns with real-world practice: a README typically contains 1–2 issues that hinder environment setup, but rarely many more. This ensures that task difficulty remains moderate for agents. Finally, in the error-type distribution in Figure 4(c), the five standard error categories are comparable in count, each around 1,600 instances, contributing to a balanced dataset. The "Other" category contains only 312 instances, which both preserves coverage completeness and discourages agents from overusing a catch-all class. Compared with other benchmarks in Table 1, our benchmark shows great advantages in evaluating the environment configuration capabilities of intelligent agents.

Table 1: Comparison of environment-configuration benchmarks.

| Benchmark | Instances | Metric | Process-level |
|---|---|---|---|
| INSTALLAMATIC(Milliken et al., 2025) | 40 | Success build | ✗ |
| EXECUTIONAGENT(Bouzenia & Pradel, 2025) | 50 | Success build & test | ✗ |
| EnvBench(Eliseeva et al.) | 994 | Success build & test, missing imports | ✗ |
| SetupBench(Vergopoulos et al., 2025) | 93 | Success build & test, | ✗ |
| EnConda-Bench | 4,201 | Success build & test, error detection and fix | ✓ |

**Data Evaluation.** Since our task instances are constructed using LLM, these generated errors may not fully reflect the real-world task conditions, so we further verify the data quality. Nevertheless, we aim to further verify whether our generated data aligns with the difficulty level of real-world environment configuration tasks and reflects human cognitive patterns. To this end, we select existing environment configuration benchmarks, whose instances are directly sourced from real-world code repositories, and establish a criterion to assess the difficulty level of both these benchmarks and our tasks. Difficulty is rated on a scale from 1 (very easy) to 5 (very hard) by human experts (see Appendix B for details). The results shown in Table 2 indicate that the difficulty distribution and average scores of our tasks closely match those of the real-world instances, demonstrating that our dataset possesses realistic applicability and high quality. To verify whether our LLM-generated errors reflect real-world distributions, we conducted a blind test with three expert engineers. We mixed 50 generated errors with 50 real-world errors collected from GitHub. As shown in Table 3, the rate at which engineers misclassified our generated errors as "Real" (54.7%) is statistically comparable to the identification rate of actual real-world errors (58.0%). This suggests that our synthetic errors are effectively indistinguishable from human-authored configuration failures.

Table 2: Difficulty scores of the benchmarks, where easy/med/hard corresponds to 1-2/3/4-5. For EnvBench and EnConda-Bench , we sample 100 task instances.

| Benchmark | Instances | Mean Score | Easy | Med | Hard |
|---|---|---|---|---|---|
| INSTALLAMATIC(Milliken et al., 2025) | 40 | 3.92 | 12 | 21 | 7 |
| EXECUTIONAGENT(Bouzenia & Pradel, 2025) | 50 | 3.85 | 16 | 26 | 8 |
| EnvBench(Eliseeva et al.) | 100 | 4.08 | 27 | 46 | 27 |
| SetupBench(Vergopoulos et al., 2025) | 93 | 3.78 | 34 | 47 | 12 |
| EnConda-Bench | 100 | 3.95 | 30 | 47 | 23 |

Table 3: Human Evaluation of Error Realism (Confusion Matrix). A high "Judged as Real" rate for LLM-generated errors indicates high realism.

| Actual Source | Judged as "Real" | Judged as "AI" |
|---|---|---|
| Real-World Errors | 58.0% | 42.0% |
| **LLM-Generated (Ours)** | **54.7%** | 45.3% |

## 3.4 EVALUATION SUITE DESIGN

After validating benchmark instances, we build an evaluation suite for environment configuration agents. Given the README and repository info, the agent plans and executes, producing a trajectory from which we extract perception (error diagnoses), feedback (repairs), and a final shell script for planning and action. Because a README may contain multiple errors, we compare the agent's predicted error types/descriptions to the gold set and report precision, recall, and F1. We then match each predicted error description and fix to the gold answer, and use GPT-4.1-mini as a judge to assess consistency and evaluate accuracy. For executability, each script runs in a Docker container on a fixed commit. A run is counted as a pass only if the environment is successfully built, the test files execute correctly, and the process exits normally. Additionally, we propose an overall data-synthesis framework that automatically generates and verifies task instances from repositories, runs agents to collect trajectories, and produces evaluations for obtaining final post- and even pre-training trajectory data (more information in Appendix C).

## 4 EXPERIMENTS

### 4.1 BASELINES

We evaluate advanced LLMs and agent frameworks (detailed setting in Appendix D). For foundation models, we include representative open- and closed-source LLMs: GPT-4.1[2], Claude-4-sonnet-20250514[3], Gemini2.5-Pro[4], and DeepSeek-V3-0324 (Liu et al., 2024a). For agent frameworks, we consider three settings: (1) **Zero-Shot**: no additional agent scaffolding. The model receives the task instance, targeted prompting for environment setup, and pointers to the repository's README and directory structure, along with evaluation configuration details (e.g., Ubuntu version), and directly produces the setup. (2) **Code Agents**: specialized software-engineering agents that benefit from tool use and planning, often trained or optimized on large SWE workloads. We evaluate OpenHands (Wang et al.) and SWE-Agent (Yang et al., 2024), both strong performers on SWE-bench-style tasks. (3) **Environment-setup Agents**: frameworks tailored to environment configuration, including INSTALLAMATIC (Milliken et al., 2025) and Repo2Run (Hu et al., 2025).

### 4.2 MAIN RESULTS

We evaluate agents on the environment setup as shown in Table 4. Zero-shot LLMs exhibit high recall but low precision in error typing (e.g., GPT-4.1: Rec. score of 90.6 vs Pre. score of 33.4, F1 score of 48.8), with weak fix suggestions and poor end-to-end success performance. This indicates broad but noisy error perception and limited agentic intelligence. In contrast, code agents significantly improve error perception and corresponding repair feedback, for instance, OpenHands + DeepSeek-V3 achieves F1 score of 58.7 with a description accuracy of 51.9. However, the action and feedback abilities are still under exploration, with fix ACC. score of 33.8 and Pass@1 score of 9.1. Environment configuration agents deliver the largest end-to-end gains, showing better capability to utilize perception and feedback with execution action. We observe that Repo2Run + Claude-4 reaches F1 score of 60.6, description accuracy of 52.2, fix accuracy of 47.3, and Pass@1 score of 22.9, underscoring the value of environment probing perception and failure handling feedback. Nonetheless, the persistent gap between description and fix accuracy, and between fix accuracy and

---

[2]https://openai.com/index/gpt-4-1/

[3]https://www.anthropic.com/news/claude-4

[4]https://deepmind.google/models/gemini/pro/

Table 4: Evaluation results across agents and LLMs. **Bold**: Optimal performance for every setting.

| Agent Capability | | Perception | | | Feedback | Feedback and Action | Planning and Action |
|---|---|---|---|---|---|---|---|
| Framework | LLM | Error type | | | Error description | Fix suggestion | Execution |
| | | Pre. | Rec. | F1 | ACC. | ACC. | Pass@1 |
| *Base Model* | | | | | | | |
| Zero-Shot | GPT-4.1 | 33.4 | **90.6** | 48.8 | 39.6 | 18.2 | 1.5 |
| | Claude-4 | **37.1** | 80.6 | **50.8** | 45.1 | **28.5** | 3.1 |
| | Gemini2.5-pro | 35.2 | 77.8 | 48.5 | **45.2** | 25.2 | 1.8 |
| | DeepSeek-V3 | 33.2 | 65.8 | 44.2 | 39.7 | 22.3 | **3.3** |
| | GPT-5-Codex | 40.5 | 88.2 | 55.5 | 49.8 | 32.4 | 6.3 |
| *Code Agent* | | | | | | | |
| SWE-Agent | GPT-4.1 | 43.7 | 83.2 | 55.3 | 49.8 | 30.7 | 7.2 |
| | Claude-4 | 46.4 | 85.6 | 58.2 | 52.4 | 34.5 | 9.4 |
| | Gemini2.5-pro | 45.1 | 92.5 | 56.8 | 50.2 | 32.2 | 7.8 |
| | DeepSeek-V3 | 41.2 | 70.3 | 51.9 | 44.5 | 27.8 | 7.4 |
| | GPT-5-Codex | 52.1 | 91.4 | 66.4 | 58.7 | 41.2 | 11.5 |
| OpenHands | GPT-4.1 | 42.5 | 72 | 53.2 | 46.0 | 29.1 | 8.5 |
| | Claude-4 | **48.0** | 87.5 | **60.1** | **54.2** | **36.1** | **10.6** |
| | Gemini2.5-pro | 45.2 | 85.2 | 57.5 | 51.8 | 32.3 | 8.5 |
| | DeepSeek-V3 | 46.7 | **93.6** | 58.7 | 51.9 | 33.8 | 9.1 |
| *Environment Configuration Agent* | | | | | | | |
| INSTALLAMATIC | GPT-4.1 | 37.5 | 70.4 | 48.9 | 45.3 | 29.1 | 5.6 |
| | Claude-4 | 41.9 | 75.3 | 53.8 | 50.7 | 34.1 | 7.9 |
| | Gemini2.5-pro | 39.0 | 72.1 | 50.6 | 47.5 | 30.8 | 6.4 |
| | DeepSeek-V3 | 40.7 | 76.8 | 53.2 | 49.3 | 32.5 | 7.1 |
| Repo2Run | GPT-4.1 | 44.2 | 72.3 | 54.8 | 48.5 | 38.6 | 14.1 |
| | Claude-4 | **49.5** | 77.5 | **60.6** | 52.2 | **47.3** | **22.9** |
| | Gemini2.5-pro | 48.6 | **79.3** | 60.1 | **54.2** | 45.6 | 17.8 |
| | DeepSeek-V3 | 46.3 | 74.2 | 56.8 | 44.6 | 41.2 | 16.2 |
| Gemini-CLI | Gemini-2.5-Pro | 48.5 | 94.2 | 64.0 | 54.6 | 38.9 | 13.2 |

Pass@1, reveals bottlenecks of the agent in translating correct feedback into robust and valid execution actions. To further benchmark the inherent difficulty of environment configuration, we conduct an experiment using the original, correct READMEs in Appendix E.5. Our process-level evaluation of agent trajectories is therefore crucial, which guides targeted improvements rather than only a pass rate, which highlights that we need to make better use of feedback information obtained from interactions with the environment, to truly enhance the execution capability.

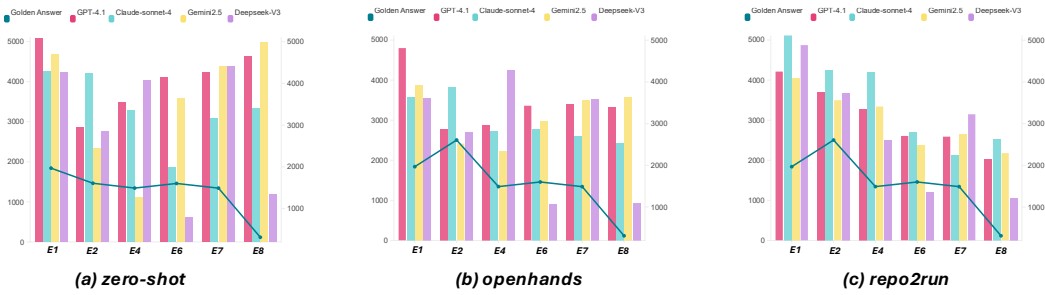

Figure 5: The statistics of the error numbers of the golden label and the model's prediction.

## 4.3 ERROR TYPE JUDGMENT ANALYSIS

For each error type, we observe that the total number of predicted errors exceeds the ground truth shown in Figure 5(see more detail in Appendix E.1), indicating a conservative strategy with stricter checks. Sensitivity to specific error types is uneven: most models tend to overpredict the E1 category. There are also model-specific differences. For example, DeepSeek-V3 predicts very few E6 cases, fewer even than the ground-truth labels, suggesting under-detection for that error type. Finally, many cases are grouped into the catch-all "other" category, making E8 the highest or second-highest category. This is undesirable as users expect precise, actionable diagnoses rather than vague

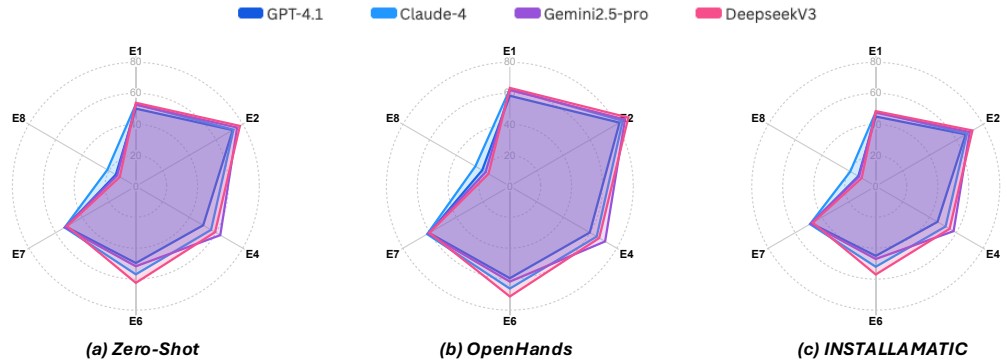

Figure 6: The illustrations of the error type judgment performance on the F1 score.

classifications. Consistent with the above, the overuse of "other" leads to a markedly low F1 score on E8 as shown in Figure 11(see more detail in Appendix E.2). Many instances that should belong to concrete types are incorrectly assigned to E8, inflating false positives for E8 and depressing recall for the true types. This hedging behavior hampers the practical value of error perception and feedback, degrading downstream planning and action. Beyond this, models show small but systematic performance differences across specific types. For example, the results show stronger detection for command usage and syntax error E2 but weaker for categories like file path error E4 that often depend on the agent's system-level understanding of the entire repository and the interaction with the environment.

## 4.4 EFFICIENCY ANALYSIS

We investigate the relationship between output tokens and performance to provide a comprehensive assessment of model efficiency. As shown in the Figure 7 (see more detail in Appendix E.3), for the accuracy of error descriptions, most models exhibit a clear upward trend as the number of output tokens increases. In contrast, for the Pass@1 metric, allocating more tokens does not consistently yield improvements. For example, zero-shot Claude-4 uses three times as many tokens as zero-shot DeepSeek-V3 yet improves performance by only about 0.2. Notably, in certain agent frameworks (e.g., Repo2Run), performance scales more favorably with larger token budgets, indicating comparatively higher efficiency.

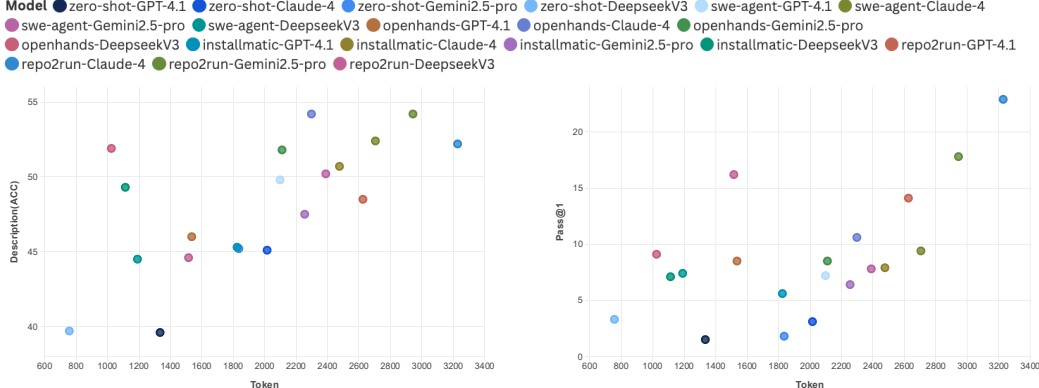

Figure 7: The illustration of the statistics of output token and model performance.

## 4.5 CASE STUDY

Beyond the above analyses, we further explore concrete cases to validate the effectiveness of our framework and support the claims. As shown in Figure 8, we observe that models sometimes cor-

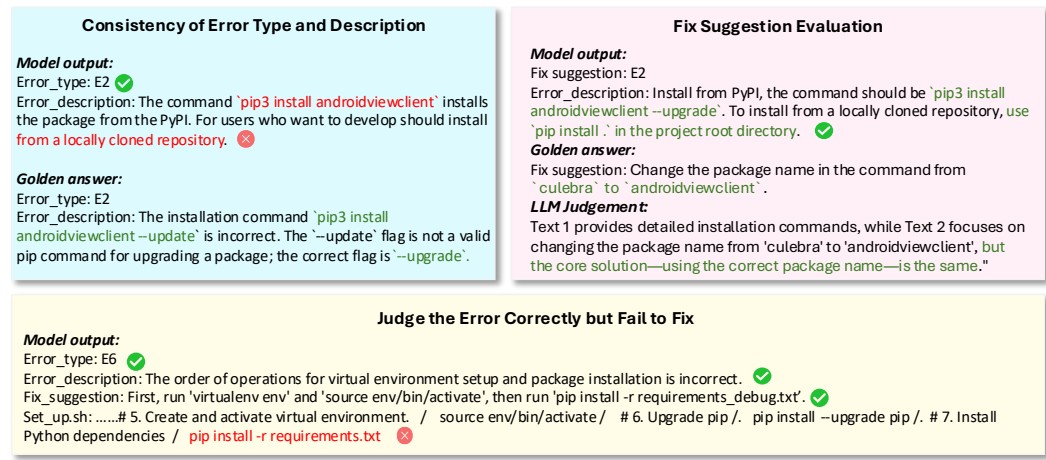

Figure 8: We select some typical cases and provide an analysis of observed phenomena.

rectly judge the error type without actually locating the error commands, which undermines subsequent feedback. We also find that some proposed fixes differ from the golden solution, but our evaluation protocol can accommodate such variability, focusing on whether the issue is resolved rather than on exact match, thereby supporting the soundness of our methodology. Finally, we observe a relatively low pass rate. One key reason is that a single README may contain multiple errors, and models often fail to fix all of them. Moreover, during execution, a model may correctly diagnose an error and suggest an appropriate fix command, but fail to apply the feedback in the shell scripts used for environment setup, or it may introduce new faults. Notably, such errors are likely to arise during multi-round agentic iteration, as extensive edits can introduce additional mistakes.

## 5 CONCLUSION

Environment configuration remains a decisive bottleneck for SWE agents. Beyond end-to-end benchmarks, we introduce a benchmark that focuses on process-level evaluation, including planning, perception-driven diagnosis, feedback-driven repair, and final actions. By injecting realistic errors into READMEs and validating their effects in Docker, the automatic data framework we propose generates scalable, high-quality task instances for evaluation and rich trajectories that enable training. We conduct experiments on advanced agents, and observe that in error perception, the agent demonstrates a certain level of capability, but tends to classify uncertain error types as "others". However, when it comes to specific repair actions, the agent shows limited performance. We attribute this limitation to the lack of effective interaction and feedback mechanisms within the agent. Although it can identify errors, it struggles to plan the feedback information and interact with the real-world environment to provide better repair actions. In the future, enhancing the agent's ability to strengthen its interaction with the environment will be an important research direction.

## ACKNOWLEDGMENTS

This work was supported in part by the New Generation Artificial Intelligence-National Science and Technology Major Project (2025ZD0123003), the National Natural Science Foundation of China Enterprise Innovation and Development Joint Fund (Artificial Intelligence Field) Key Support Projects (U25B2072), the Basic Research Fund of Shenzhen City JCYJ20240813112009013, and The Major Key Project of PCL (Grant No. PCL2025A17).

## ETHICS STATEMENT

We introduce a novel benchmark, `EnConda-Bench` , incorporating a thorough description of repository collection, error synthesis, data validation, and filtering. We emphasize that the dataset's

creation adheres strictly to ethical guidelines. We make sure that all repositories we use comply with their respective licenses. Great care has been taken to uphold ethical standards in the dataset, employing anonymization, desensitization, and data cleaning. The samples pose no risk to public welfare. For all data sourced from these websites, we obtain permission for data usage. Hence, the innovative research directions and tasks proposed are ethically harmless to society.

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

## A  DATA CONSTRUCTION

### A.1  REPOSITORY SELECTION

We focused on Python repositories given their dominance in the ecosystem (over 21M repositories on GitHub). To ensure benchmark quality, we conduct:

1. Selection Criteria: We adopted the criteria of 10+ stars, 1000+ commits, and 10+ closed issues based on established practices in software engineering research. These metrics serve as proxies for repository maturity, active maintenance, and community relevance, ensuring we benchmark on meaningful projects rather than toy examples.

2. Filtering Pipeline: We strictly filtered for repositories with precise READMEs located in the root directory to minimize ambiguity.

3. Rigorous Verification: The "manual checking" involved *3 expert engineers. They manually attempted to set up the environment following the original READMEs. Any repository that failed to build or had ambiguous instructions was strictly removed. This ensures that every Ground Truth in our benchmark is guaranteed to be executable.

Beyond the filtering criteria outlined in Section 3.2, we selected repositories primarily from existing, human-validated, and pre-filtered benchmarks to ensure the originals are reliably buildable. This choice is crucial because our error synthesis procedure treats the unmodified README as the ground truth; consequently, the README must be correct and actionable. To that end, we engaged professional annotators to perform manual environment setup, removing repositories whose READMEs were themselves erroneous or whose dependencies were incomplete, thereby preserving high-quality baselines. We further verified that all selected repositories carry licenses permitting non-restrictive research use: most adopt permissive licenses (e.g., BSD, MIT, Apache), while the remainder fall under copyleft licenses (e.g., GPL), which are compatible with our intended research

scenarios. We also reviewed repositories under custom licenses and confirmed their suitability for the uses contemplated in this study. A promising direction for future work is to broaden repository coverage to include codebases written in a wider range of programming languages, extending the evaluation to environment configuration tasks across more language ecosystems.

## A.2 Error Type Definition

**Taxonomy of Common Environment-Configuration Errors**    To ensure our error taxonomy covers real-world failure modes, we adopted a multi-source derivation process. First, we surveyed existing empirical studies on configuration errors to identify historical patterns. Second, we mined common issues from developer communities like StackOverflow and GitHub. Finally, we refined the taxonomy through interviews with five expert engineers (10+ years of experience), merging correlated types and removing non-typical ones. This process resulted in the six defined error types, with an "Other" category to capture long-tail distributions.

Guided by failure modes frequently encountered when configuring execution environments from software repositories, we define a six-type taxonomy intended to cover the vast majority of practical issues while maintaining clear, operational boundaries between categories. The scheme is designed to support consequent error generation, and the set is comprehensive for routine evaluation and error synthesis.

Table 5: Error taxonomy for repository environment configuration. Identifiers are retained from our internal schema and are non-contiguous by design.

| ID | Name | Definition |
|----|------|------------|
| E1 | Dependency Installation Error | Errors related to system or Python dependency installation steps, including missing dependencies, unnecessary dependencies, or version errors. |
| E2 | Command Usage or Syntax Error | Errors caused by incorrect commands, invalid parameters, or improper syntax causing execution failure. |
| E4 | File Path or Missing File Error | Errors where dependency file paths are incorrect or referenced files do not exist. |
| E6 | Logical Order Error | Errors caused by incorrect execution order of installation steps, such as installing pip dependencies before creating a virtual environment. |
| E7 | Version Compatibility Error | Errors caused by unspecified Python or dependency versions, version conflicts, or incompatibilities. |
| E8 | Other Miscellaneous Errors | Other uncategorized errors, such as messy formatting, missing critical explanations, or unclear descriptions. |

**Examples**

- E1  Typical symptoms include package-not-found errors (e.g., 404 on indexes or channels), missing system libraries (e.g., OpenSSL, GCC toolchains), or failing installers (apt/conda/pip).

- E2  Often manifests as immediate termination with usage messages or exit code 2, invalid or deprecated flags, shell quoting/escaping errors, or invoking commands from the wrong working directory.

- E4  Common signals are "No such file or directory," misspelled filenames, incorrect relative paths, or reliance on artifacts not checked into version control.

- E6  Symptoms include installing into an inactive environment, failing to source activation scripts before use, or attempting builds before installing toolchains.

- E7 Presents as solver conflicts, runtime ImportErrors due to ABI/GLIBC/CUDA mismatches, or subtle behavior differences across Python or library minor versions.

- E8 Catch-all for issues outside the above scopes, such as incomplete or ambiguous instructions, inconsistent naming, or extraneous formatting that obscures required steps; curators should use this sparingly and prefer specific categories when feasible.

## A.3 ERROR GENERATION

We use Claude-4-sonnet and Gemini 2.5-pro together to generate errors. We input the original README file, the desired types of errors to insert, the number of errors per README, the desired number of output README files, and a list of error type definitions, and then have them output the following:

- Erroneous README (Markdown or RST file)

- List (JSON format):

  - Readme id

  - Error type

  - Error description (natural language)

  - Candidate fix suggestions (operational tips)

  - ground truth of fix answer (golden answer)

Here is the instruction for error generation:

---

*Prompt for Error Synthesis*

```
You are a professional environment setup engineer and the README text
modifier.
You receive the following inputs:

- A correct README markdown file located at path: {readme_path}.
- A list of error types to inject, chosen from the following:

{error_types_str}
- Number of errors to insert per README: {errors_per_readme}.
- Number of distinct erroneous README markdown files to output:
{num_readmes}.

Task:

Please generate {num_readmes} distinct erroneous README markdown files
by minimally modifying the original README. Keep as much of the
original README the same as possible, and only inject errors
by changing or adding a very small number of sentences (usually 1 or 2)
to introduce the requested error types.

For each generated erroneous README file, output the following parts:

1. The full erroneous README markdown text, preserving all original
formatting and content except the minimal injected errors.
2. A JSON metadata block describing all inserted errors, with the
structure:
```

---

*Prompt for Error Synthesis*

```json
{{
  "readme": "readme_{{index}}",
  "errors": [
    {{
      "error_type": "<error type code e.g., E1>",
      "error_description": "<brief description of the introduced error
      in this README>",
      "correction_candidates": [
        "<candidate fix #1>",
        "<candidate fix #2>"
      ],
      "golden_answer": "<precise correction to fix the error>"
    }}
  ]
}}
```

Please output results for all {num_readmes} files in this format,
separated by a line of exactly three dashes (---):
---
# Erroneous README {{index}}
<full markdown text>

---
```json
{{
  "readme": "readme_{{index}}",
  "errors": [
    ...

  ]
}}
```
---
Additional notes:
- Ensure JSON blocks are enclosed exactly by triple backticks
and labeled as json.
- Maintain overall readability and realistic style.
- Output the markdown and JSON blocks exactly as specified, with no
extra text or
commentary.
- The original README content, not injected errors, must remain
unchanged.

End of instructions.

## A.4 AUTOMATIC VALIDATION

After generating the erroneous README file, as in the formal evaluation process, we used GPT-4.1-mini to generate a shell script based on the given README and the directory structure of the repository. This script was then run within Docker to verify whether the environment could be successfully built. We instructed the model to strictly follow the instructions in the README when generating the script, without making any modifications or corrections, to ensure accurate evaluation results. The specific prompt used is as follows:

*Prompt for Generating Shell for Validation*

```
You are given the README of a Python project and the directory
structure of this repository. Please output ONLY one complete
bash shell script that automates environment setup for this
project on Ubuntu 22.04. Do not include any explanations or
markdown, just the script content.

Requirements for the script:
- Create and activate a new Python virtual environment using Miniconda
inside the project directory (e.g., ./env or ./venv name).
- Install any required system packages and Python packages inferred
from the README.
- Install Python dependencies (infer from README; if requirements.txt/
pyproject.toml is expected, handle both).
- Install the project in editable mode (pip install -e .)
if applicable.
- You must run the test suite to verify setup (pytest, or whatever
is indicated/inferred).
- Start with a shebang (#!/usr/bin/env bash) and use
'set -euo pipefail'.
```

*Prompt for Generating Shell for Validation*

```
Attention: You don't have to ensure that the generated Shell script
will definitely configure successfully, but **make sure that the
Shell script is totally consistent with the contents of the
README**. Do not make any changes!
Assume you execute all commands from the project root directory.
```

## A.5 LLM AS JUDGE TO VALIDATE

We further annotate each error using GPT-4.1-mini to ensure that the final generated errors are valid. We input the error's README, a JSON file containing the error annotation, the error definition, and the basic Dockerfile design for our environment configuration. We ask GPT-4.1-mini to check the following: (1) Is the error type classification accurate? If not, suggest a corrected type. (2) Is this error described in the README? (3) Is this error valid? We consider an error valid only if it truly prevents a step in the environment configuration from succeeding, and its fix allows the configuration to execute correctly. (4) Is the standard solution for this error correct? After this screening, we discard all READMEs corresponding to invalid errors. The specific prompt is as follows:

*Prompt for LLM as Judge to Filter the Valid Error README*

```
SYSTEM_PROMPT
You are a strict checker for environment-setup. Output only N lines of
minified JSON objects, one per error, exactly matching the schema
below. No explanations, no prose, no code fences, no blank lines,
no trailing commas.

DEVELOPER_PROMPT
Task:
For each error in errors_json.errors (in order), decide:
- error_type_judgment: does error_type match error_description per
definitions? If false, set error_type_modify to the single best-
matching type; else "".
- error_readme_cr: does the README text actually contain the erroneous
content (wrong command/flag/version/path/module, wrong venv order)
or a required omission that would be needed to succeed?
- answer_judgment: would golden_answer remove the root cause of this
error?
```

---

*Prompt for LLM as Judge to Filter the Valid Error README*

```
- error_valid: will this error break setup until fixed under the given
environment
assumptions and success criteria (install command errors, dependency
resolution
conflict, ImportError of a required library, unusable environment due
to venv misuse)? False for cosmetic/optional issues or if uncertain.
Use only:
- readme_text
- error_definitions
- errors_json
- environment assumptions/success criteria
Be conservative if context is missing. Preserve input order.

Output schema (one line per error)
{"id":"<readme_id>#<zero_based_index>",
"error_type_judgment":true|false,"error_type_modify":"E? or empty
string",
"error_readme_cr":true|false,"answer_judgment":true|false,"error_valid
":true|false}
Rules
- error_type_judgment: compare strictly with provided definitions;
if multiple types fit, choose the most specific root-cause category.
- error_readme_cr: true only if the README substantiates the error;
for omissions,
true only if the missing step/dependency is clearly required to
succeed.
- answer_judgment: golden_answer must directly fix the error; being in
correction_candidates is not sufficient.
- error_valid: true for invalid flags/options, incompatible/blocked
pins, missing essential deps, wrong commands/paths/modules,
critical venv misuse/order; false for stylistic advice or
speculative failures.

Expected model output (one line per error, in order):
{"id":"<readme_id>#0","error_type_judgment":true,
"error_type_modify":"","error_readme_cr":true,"answer_judgment":
true,"error_valid":true}
```

## B  DIFFICULTY RATE

Because our task instances incorporate error injection into existing README files using LLM, we wanted to verify whether these generated task instances reflect the challenges encountered in real-world environment configuration, thus possessing characteristics similar to real-world environment setup tasks. Therefore, we selected several benchmarks that collect task instances directly from real-world code repositories, including INSTALLAMATIC Bench, ExecutionAgent Bench, EnvBench, and SetupBench, and compared their difficulty scores with our benchmark. For EnvBench and our benchmark, we sampled 100 task instances each; for the other benchmarks, we scored all available instances. By analyzing the distribution of difficulty scores, we can assess whether the methods used in our benchmark align with those used in real-world task instances. Specifically, we used a 1-5 scale, where 1 is very easy, 2 is easy, 3 is moderate, 4 is difficult, and 5 is very difficult. We invited professional annotators to score the selected task instances, considering factors such as the clarity and completeness of the instructions in the README, whether the commands execute directly, whether additional files or pages need to be consulted, and the number of dependencies that need to be considered.

## C  TRAJECTORY TRAINING DATA GENRATION FRAMEWORK

We have automated the process of generating environment configuration task instances and designed a comprehensive evaluation suite that allows agents to execute these task instances, capture

Figure 9: The overall data synthesis framework of trajectory training data generation.

their execution trajectories, and perform evaluations. Therefore, we can build a complete synthetic data framework based on this to generate synthetic trajectory data representing both successful and failed agent executions of these environment configuration tasks. This can efficiently produce large amounts of trajectory data for model fine-tuning or large-scale pre-training, provided that a sufficient quantity of high-quality original repository data is available. The specific data generation process is illustrated in the Figure 9.

# D    EXPERIMENT SETTINGS

When the agent is tasked with configuring the environment, we provide instructions including the repository directory information, README information, and the basic environment requirements. We also outline a feasible workflow for the agent to follow, ensuring that the entire environment configuration adheres to the specified standards. We require the agent to explicitly identify any errors during execution, and to perform unit tests after completing the environment configuration to verify its success. The specific prompt is as follows:

*Prompt for Agent to Execute the Environment Configuration*

You are an expert Python environment setup assistant. Your task is to analyze README files, detect potential errors in environment setup instructions, and provide comprehensive solutions.

Given a README file, you should:

1. **Error Detection and Analysis**: Carefully analyze the README for potential errors in environment setup instructions, including:
   - E1: Dependency Installation Error (missing dependencies, unnecessary dependencies, or version errors)
   - E2: Command Usage or Syntax Error (incorrect commands, invalid parameters, or improper syntax)
   - E4: File Path or Missing File Error (incorrect dependency file paths or referenced files that do not exist)
   - E6: Logical Order Error (incorrect execution order of installation steps)
   - E7: Version Compatibility Error (unspecified Python or dependency versions, version conflicts, or incompatibilities)
   - E8: Other Miscellaneous Errors (messy formatting, missing critical explanationsand or unclear descriptions)

2. **Error Analysis Output**: First, output a JSON object containing your error analysis with the following structure:
```json
{{
  "detected_errors": [
    {{
      "error_type": "E1|E2|E4|E6|E7|E8",
      "error_description": "Detailed description of the error found",
      "fix_suggestion": "Specific suggestion on how to fix this error"
    }}
  ]
}}
```
3. **Environment Setup Script**: After the error analysis, create a comprehensive shell script that:
   - Fixes all detected errors
   - Sets up the environment correctly
   - Handles common Python environment setup patterns (pip, conda, poetry, etc.)
   - Includes error handling and verification steps

Your response should contain:
1. The JSON error analysis (wrapped in ```json code blocks)
2. The corrected shell script (wrapped in ```bash code blocks)

Technical requirements:
- Always start by examining the repository structure to locate the dependency
- Check for Python version requirements and use pyenv for version management
- Identify the dependency manager (pip, Poetry, etc.) and use it appropriately

---

*Prompt for Agent to Execute the Environment Configuration*

```
- Handle system-level dependencies with apt-get
- Ensure proper virtual environment setup
- Include verification steps to confirm successful installation
- Use non-interactive commands (e.g., 'apt-get install -y')
- Install from local repository, not PyPI packages

## Repository Structure:
{file_structure}

## README Content:
{readme_content}

Please analyze the above README file and provide your response
in the specific format.
```

---

# E ADDITIONAL EXPERIMENT RESULTS

## E.1 ADDITIONAL RESULTS OF THE STATISTICS OF ERROR JUDGMENT NUMBER

We counted the number of errors in the agent's output and compared it with the number of errors in the correct answer. The complete results are shown in Figure 10. Overall, the results still indicate that the agent tends to produce more incorrect judgments than the correct answer, employing a more stringent strategy.

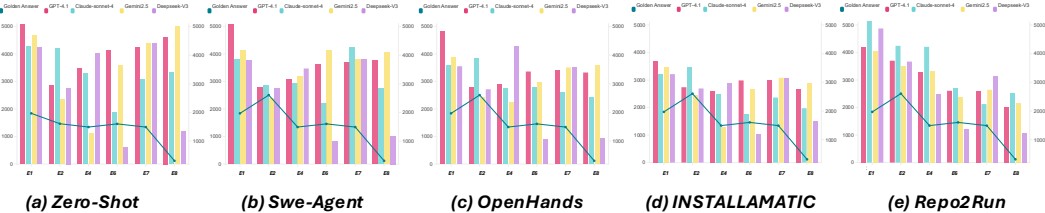

Figure 10: The complete results of the statistics of error judgment number.

## E.2 ADDITIONAL RESULTS OF ERROR TYPE JUDGMENT PERFORMANCE

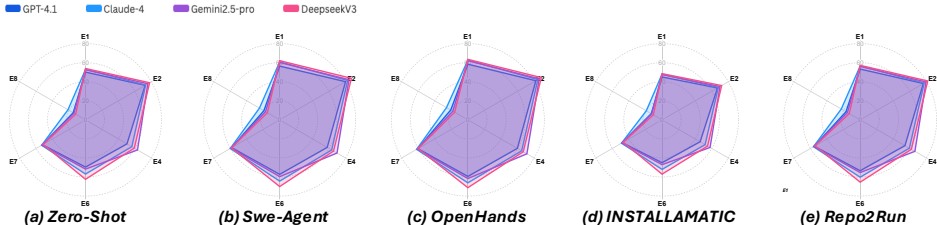

Figure 11: The illustrations of the error type judgment performance on the F1 score.

## E.3 ADDITIONAL RESULTS OF THE STATISTICS OF MODEL OUTPUT TOKEN

We analyzed the relationship between the number of tokens generated by the agent model and its performance across various metrics, in order to further evaluate the efficiency of different models. The complete statistical results for these different metrics are shown here.

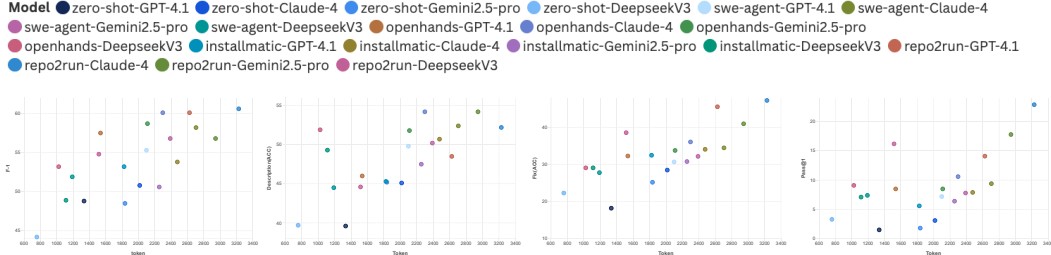

Figure 12: The complete results of the statistics of model output token.

### E.4 IMPACT OF ERROR COUNT (DIFFICULTY).

We analyzed model performance across different difficulty levels based on the number of injected errors. As presented in Table 6, the F1-score exhibits an "inverted-U" trend, typically peaking at Level 5 or 7. At lower levels (L1-L3), models tend to hallucinate additional errors, resulting in low precision. Conversely, at high levels (L10), while precision improves, recall drops significantly as models struggle to identify every error in highly defective files. This confirms that moderate complexity (L5-L7) represents the "sweet spot" for current agents.

Table 6: Ablation Study on Error Difficulty Levels. Metrics are reported as Precision / Recall / F1-Score.

| Model | Level 1 | Level 3 | Level 5 | Level 7 | Level 10 |
|---|---|---|---|---|---|
| GPT-4.1 | 27.6/40.2/32.7 | 38.7/46.4/42.2 | 74.2/44.6/**55.7** | 84.2/35.0/49.4 | 88.4/20.6/33.4 |
| Claude-4-sonnet | 29.3/41.6/34.4 | 47.8/28.8/36.0 | 74.6/38.5/**50.8** | 79.8/33.9/47.6 | 85.7/25.0/38.7 |
| Gemini-2.5-pro | 28.0/42.9/33.9 | 47.6/28.1/35.4 | 71.8/35.0/47.1 | 76.4/47.2/**58.3** | 67.2/23.4/34.7 |
| DeepSeek-V3 | 27.5/34.3/30.5 | 43.0/20.5/27.8 | 64.7/32.3/**43.1** | 71.0/29.6/41.8 | 71.4/26.3/38.4 |

### E.5 ADDITIONAL EXPERIMENTS WITH CORRECT README

To further benchmark the inherent difficulty of environment configuration, we conduct an additional experiment using the original, correct READMEs (i.e., without injected errors) under identical settings. As shown in Table 7, while the End-to-End Pass@1 rates on correct READMEs are slightly higher than those on the erroneous ones, the overall success rate remains surprisingly low. This observation confirms that automated environment configuration is a non-trivial task even when provided with correct instructions.

Table 7: Comparison of model performance on Erroneous vs. Correct READMEs. The results show that while correct READMEs yield slightly higher success rates, the task remains challenging. Note that process-level metrics (Pre/Rec/F1) are not applicable to correct READMEs as they contain no ground-truth errors.

| Framework | LLM | Error Type Judgment | | | Error Desc. | Fix Suggestion | End-to-End | Correct README |
|---|---|---|---|---|---|---|---|---|
| | | Pre | Rec | F-1 | ACC | ACC | Pass@1 | Pass@1 |
| *Base Model* | | | | | | | | |
| Zero-shot | GPT-4.1 | 33.4 | 90.6 | 48.8 | 39.6 | 18.2 | 1.5 | 4.5 |
| | DeepSeek-V3 | 33.2 | 65.8 | 44.2 | 39.7 | 22.3 | 3.3 | 6.3 |
| *Code Agent* | | | | | | | | |
| SWE-agent | GPT-4.1 | 43.7 | 83.2 | 55.3 | 49.8 | 30.7 | 7.2 | 9.2 |
| | DeepSeek-V3 | 41.2 | 70.3 | 51.9 | 44.5 | 27.8 | 7.4 | 11.3 |

## F  THE USE OF LARGE LANGUAGE MODELS (LLMs)

In accordance with the official policy on the use of large language models, we used LLMs solely as general-purpose assistive tools for grammar checking and minor wording refinement during manuscript preparation. All LLM-suggested edits were manually reviewed and selectively accepted by the authors. Our usage complies with the official requirements, and we disclose it here.

