# OpenReview forum: "Process-Level Trajectory Evaluation for Environment Configuration in Software Engineering Agents"
_ICLR.cc/2026/Conference — ICLR 2026 Poster_

### Official Review · Reviewer_pDzU · 2025-10-26

**Soundness:** 3
**Presentation:** 3
**Contribution:** 3
**Rating:** 6
**Confidence:** 4

**Summary:**

This paper introduces a benchmark, EnConda-Bench, for evaluating LLM-based agents on environment configuration tasks in software engineering. The key innovation is providing process-level trajectory assessment across four capabilities: environment setup-planning, perception-driven error diagnosis, feedback-driven repair, and action execution. This work constructs 4,201 tasks from 323 GitHub repositories. Their evaluation across multiple LLMs and agent frameworks reveals that. while agents can localize errors reasonably well (F1 ~60%), they struggle to translate feedback into effective corrections.

**Strengths:**

1. This paper propose a new benchmark with process-level evaluation for environment configuration.

2. From repository selection to filtering and validation, the multi-stage dataset construction pipeline demonstrates rigor.

3. The paper clearly articulates the problem (environment configuration bottleneck), motivation (limitations of end-to-end metrics), and solution (process-level evaluation with synthetic errors).

**Weaknesses:**

**1. Limitations In Language Coverage**: The benchmark focuses exclusively on Python repositories. Given that environment configuration challenges exist across all programming languages, this can limit the generalizability of the evaluation and conclusions.

**2. Limitations In Synthetic Error Validity**: While the authors validate that injected errors cause failures, there's insufficient evidence that these errors represent the *distribution* of real-world configuration problems. The difficulty comparison (Table 2) shows similar mean scores among different benchmarks, but doesn't validate whether the *types* of errors match real-world distributions.

**3. Limitations In Evaluation Metrics:** Pass@1 metric doesn't account for partial progress (e.g., fixing 1 of 2 errors).

**4. Limitations In Analysis:** The specialized environment agents (e.g., Repo2Run) are evaluated but not deeply analyzed for why they perform better. There are also no ablation studies examining which agent design choices matter most.

**5. Limitations In Data Construction Transparency**: The repository selection criteria (10+ stars, 1000+ commits, 10+ issues) seem arbitrary, while no justification are provided. Also, the process of "manual checking" is mentioned but not detailed (e.g., how many annotators? what was the failure rate?).

**6. Limitations In Paper Presentation:** Some figures are hard to read. For example, all the scatter points and X-axis labels and titles are very hard to read in Figure 7, and model names are also very hard to read in Figure 5.

**Questions:**

My questions are following several aspects mentioned in weakness:

- How did you validate that your synthetic error distribution matches real-world configuration problems?
- How did you take partial credit into consideration, as it cannot be effectively assessed by Pass@1?
- Can you provide more detailed analysis of *why* Repo2Run performs better? Is it the dual-environment architecture, the rollback mechanism, or something else?
- Through case study, what are the common failure modes or patterns? What are their distributions? What are the implications for future improvements?
- Can you give more detailed explanation for selection, filtering, and validation criteria in dataset construction?
- Can you revise the figures to make them clearer and more readable?

---

> ### Author Response · Authors · 2025-11-22
> **Response to Reviewer pDzU (Part 1/3)**
>
> We sincerely thank you for your insightful comments and constructive feedback, and we appreciate the time you took to review our paper. We hope our response below addresses your concerns.
>
> ## Response to Weaknesses:
>
> **Response:**
>
> > Limitations In Language Coverage: The benchmark focuses exclusively on Python repositories. Given that environment configuration challenges exist across all programming languages, this can limit the generalizability of the evaluation and conclusions.
>
> Thank you for this suggestion. We selected Python as the primary language for this study due to its predominance in the software ecosystem. As of June 1, 2025, there are over **21 million** Python repositories on GitHub [1], making Python environment configuration a critical and representative task. Existing research in this domain, such as Repo2Run [2] and Installamatic [3], also focuses specifically on Python, highlighting its importance as a research testbed. In addition, our team and the expert engineers involved in the annotation process possess deep expertise in the Python ecosystem. This domain knowledge was crucial for ensuring the validity of data construction and verification. Also, we agree that expanding to multi-language support would further enhance the benchmark's generalizability. We plan to incorporate other languages in future work.
>
> *[1] Pavel A, Mikhail I, Valeev A, et al. SWE-MERA: A Dynamic Benchmark for Agenticly Evaluating Large Language Models on Software Engineering Tasks[C]//Proceedings of the 2025 Conference on Empirical Methods in Natural Language Processing: System Demonstrations. 2025: 440-452.*
>
> *[2] Hu R, Peng C, Xu J, et al. Repo2run: Automated building executable environment for code repository at scale[C]//The Thirty-ninth Annual Conference on Neural Information Processing Systems. 2025.*
>
> *[3] Milliken L, Kang S, Yoo S. Beyond pip install: Evaluating llm agents for the automated installation of python projects[C]//2025 IEEE International Conference on Software Analysis, Evolution and Reengineering (SANER). IEEE, 2025: 1-11.*
>
> > Limitations In Synthetic Error Validity: While the authors validate that injected errors cause failures, there's insufficient evidence that these errors represent the distribution of real-world configuration problems. The difficulty comparison (Table 2) shows similar mean scores among different benchmarks, but doesn't validate whether the types of errors match real-world distributions.
>
> **Response:**
>
> First, when using LLM for error injection, we strive to **ensure the diversity and effectiveness of errors, aligning them with real-world distributions**. During error injection, we first generate 4-8 erroneous README documents for each correct README, with each erroneous document containing two errors. During generation, we provide all error types to the LLM for autonomous selection and generation, rather than directly specifying only two errors per category. Furthermore, after validating the error effectiveness, we merge and split the errors in the documents, generating README documents containing 1-16 errors, further enriching the diversity. It's worth noting that we did not limit the number of errors generated for each error category, but the final statistical results (see Figure 4) show a **relatively balanced number of errors across different types**, demonstrating the effectiveness of this generation strategy.
>
> In addition, to verify whether the LLM-injected errors conform to real-world distributions, we conducted a comparative experiment in Table 2. We simultaneously challenged and scored our benchmark against unmodified repositories directly obtained from the real world. We found that the difficulty distribution was similar, proving that the data used in our benchmark conforms to real-world patterns. Furthermore, to specifically **investigate whether errors generated by LLM accurately reflect real-world error patterns**,** we supplemented the study with a new human evaluation experiment. We randomly selected 50 generated errors from the benchmark and collected 50 real-world configuration errors of the corresponding error types from GitHub. These were then combined and scored by three human engineers, who selected which errors were generated by LLM and which were real-world errors. The results are as follows:
>
> **Table: Human Evaluation of Error Realism (Confusion Matrix)**
> | Actual Source | Judged as "Real" | Judged as "AI" |
> | :--- | :---: | :---: |
> | **Real-World Errors** | 58.0% | 42.0% |
> | **LLM-Generated (Ours)** | **54.7%** | 45.3% |
>
> The results show that engineers identified our generated errors as "Real" at a rate (54.7%) very similar to actual human errors (58.0%), suggesting **our errors are indistinguishable from real-world ones**. We have added this experimental result in the revised paper in Page 6, Line 308-312 and Page 7, Line 324-330.

---

> ### Author Response · Authors · 2025-11-22
> **Response to Reviewer pDzU (Part 2/3)**
>
> > Limitations In Evaluation Metrics: Pass@1 metric doesn't account for partial progress (e.g., fixing 1 of 2 errors).
>
> **Response:**
>
> We agree that Pass@1 is insufficient on its own. That is exactly why our evaluation suite explicitly accounts for partial progress through the **"Fix Suggestion Accuracy"** metric. This metric measures the consistency between the agent's proposed fixes and the golden answers. It calculates the proportion of correctly fixed errors out of the total number of errors, effectively quantifying partial success even if the end-to-end build fails. Moreover, Pass@1 remains the gold standard for functional success (indispensable for configuration tasks), our contribution lies in augmenting it with process-level metrics. This allows us to diagnose *why* an agent fails—e.g., it may correctly identify errors (high detection accuracy) but fail to fix all of them (lower fix accuracy), leading to a failed Pass@1.
>
> > Limitations In Analysis: The specialized environment agents (e.g., Repo2Run) are evaluated but not deeply analyzed for why they perform better. There are also no ablation studies examining which agent design choices matter most.
>
> **Response:**
>
> Thank you for your suggestions! Since our **primary motivation was to establish a benchmark and related evaluation suite capable of evaluating the agent execution process** for software engineering environment configuration tasks, we did not conduct further ablation experiments on the architecture design of specific agents. In the case study 8 (page 10), we presented **some cases and analyzed some typical failure phenomena** that occurred during the agent's execution.
>
> For some specialized environment agents, we further supplemented the analysis with the reasons for their superior performance. We noticed that their better performance is often due to their **architecture design demonstrating better observation capabilities and feedback-based remediation strategies** for the specific execution process of environment configuration tasks. For example, in repo2run, due to its **dual-environment dynamic monitoring strategy**, it has better remediation solutions for feedback from environment configuration tasks. We will further refinine these experimental analyses, including incorporating ablation experiments for different architecture designs into future work.
>
> > Limitations In Data Construction Transparency: The repository selection criteria (10+ stars, 1000+ commits, 10+ issues) seem arbitrary, while no justification are provided. Also, the process of "manual checking" is mentioned but not detailed (e.g., how many annotators? what was the failure rate?).
>
> We sincerely apologize for any inconvenience caused. We have clarified the data construction pipeline to improve transparency, and the relevant details have been improved in the revised version of the paper (see Page 12 Line 630-642).
> 1.  **Selection Criteria:** We adopted the criteria of **10+ stars, 1000+ commits, and 10+ closed issues** based on established practices in software engineering research. These metrics serve as proxies for repository maturity, active maintenance, and community relevance, ensuring we benchmark on meaningful projects rather than toy examples.
> 2.  **Filtering Pipeline:** We strictly filtered for repositories with precise READMEs located in the root directory to minimize ambiguity.
> 3.  **Rigorous Verification:** The "manual checking" involved **3 expert engineers**. They manually attempted to set up the environment following the original READMEs. Any repository that failed to build or had ambiguous instructions was strictly removed. This ensures that every "Ground Truth" in our benchmark is guaranteed to be executable.
>
> > Limitations In Paper Presentation: Some figures are hard to read. For example, all the scatter points and X-axis labels and titles are very hard to read in Figure 7, and model names are also very hard to read in Figure 5.
>
> **Response**：
>
> We sincerely apologize for the difficulty caused by the small figure sizes. Due to page limits, we initially attempted to compress as much information as possible into the main text instead of appendix.
>
> In the revised manuscript, we have **enlarged Figures 4 (Page 5, Line 251-269), 5 (Page 8,Line 410-421), 6 (Page 9, Line 432-445) snd 7 (Page 9, Line 466-480)** to ensure they are clearly readable. To accommodate these changes within the page limit, we have selected the most representative sub-figures for the main text and moved the complete set of figures to the Appendix (Page 20-21, Line 1043-1090). We are committed to further optimizing the layout to improve the reading experience.

---

> ### Author Response · Authors · 2025-11-22
> **Response to Reviewer pDzU (Part 3/3)**
>
> ## Response to Questions:
>
> > **Q1: How did you validate that your synthetic error distribution matches real-world configuration problems?**
>
> **Response:**
> We validated this through two methods:
> 1.  **Difficulty Alignment:** We compared the difficulty distribution (Challenge Score) of our benchmark against unmodified real-world repositories and found them to be consistent.
> 2.  **Human Evaluation:** We conducted a "Turing Test" where engineers could not distinguish our generated errors from real ones (see the Table in **Response to Weakness 2**).
>
> > **Q2: How did you take partial credit into consideration?**
>
> **Response:**
> As detailed in **Response to Weakness 3**, we utilize **"Fix Suggestion Accuracy"** as the metric for partial credit. It calculates the percentage of correctly resolved errors within a repository, allowing us to credit agents that fix some, but not all, issues.
>
> > **Q3: Can you provide more detailed analysis of why Repo2Run performs better?**
>
> **Response:**
> Yes. As detailed in **Response to Weakness 4**, our analysis attributes Repo2Run's success to its **dual-environment architecture**, which enables superior observation of runtime feedback and more precise rollback/repair mechanisms compared to general-purpose coding agents.
>
> > **Q4: Through case study, what are the common failure modes?**
>
> **Response:**
> Our case studies (Page 10) reveal that common failure modes include:
> 1.  **Over-correction:** Agents identifying non-existent errors (hallucination).
> 2.  **Imprecise Localization:** Correctly identifying the error type but failing to locate the specific command line.
> 3.  **Cascading Failures:** Fixing one error but introducing a new dependency conflict due to lack of global context.
>
> > **Q5: Can you give more detailed explanation for selection... in dataset construction?**
>
> **Response:**
> Please refer to **Response to Weakness 5** for the detailed criteria and the rigorous 3-person manual verification process.
>
> > **Q6: Can you revise the figures?**
>
> **Response:**
> Yes, we have enlarged Figures 4, 5, 6, and 7 and optimized the layout in the revised manuscript.
>
> ---
>
> All additional experiments and clarifications have been updated in the revised manuscript and are **highlighted in blue**, including human evaluation of error realism (paper in Page 6, Line 308-312 and Page 7, Line 324-330), clarification for repository selection (Page 12, Line 630-642) and better presentation of figures (Figures 4 (Page 5, Line 251-269), 5 (Page 8,Line 410-421), 6 (Page 9, Line 432-445) snd 7 (Page 9, Line 466-480), Appendix (Page 20-21, Line 1043-1090)). We once again thank you for your review and remain open to further discussion to improve our work.

---

> ### Author Response · Authors · 2025-11-28
> **Friendly Reminder for Discussion**
>
> As the rebuttal and discussion phases are drawing to an end, we wanted to gently follow up as we haven't received your feedback yet. If you have any questions or further comments regarding our responses, please feel free to share them. We are committed to addressing them during the remaining discussion period!

---

### Official Review · Reviewer_qXby · 2025-11-01

**Soundness:** 2
**Presentation:** 3
**Contribution:** 3
**Rating:** 4
**Confidence:** 4

**Summary:**

In this paper, the authors propose a framework for analyzing process-level trajectories of LLM agents in environment configuration. This moves evaluation beyond simple pass/fail build outcomes to diagnose where and why an agent fails. The dataset is created by injecting errors (six defined categories) into valid README files followed by automatic Docker-based validation. However, this methodological strength also possibly limits the benchmark's real-world applicability (discussed more in Weaknesses section).

**Strengths:**

1. This work addresses a critcal yet unexplored bottleneck, moving from end-to-end pass/fail metrics to process-level trajectory assessment. This is useful to extract actionalble feedback that is useful for agent designers.

2. The decomposition of evaluation into the perception, feedback, and action provides fine-grained diagnostics beyond aggregate success metrics.

**Weaknesses:**

1. The six error types chosen are said to be "guided by failure modes frequently encountered in practice", but no citation or prior empirical study is provided to substantiate this taxonomy. Without grounding in developer-observed data, it is unclear whether these six types cover real-world failure modes, or simply reflect intuitive assumptions.

2. Each erroneous README is created by injecting two errors per file. This raises two concerns: (i) since both synthesis and evaluation rely on LLM behavior, the resulting benchmark may reflect model-specific phrasing or error styles, rather than human-authored configuration errors; (ii) generating only two errors per category constrains diversity, a stronger design would sample multiple error candidates and retain a stratified subset verified by humans.

3. The results lack ablation on error difficulty levels, number of injected errors, and impact of Docker environment variations. These would strengthen the claim that process-level evaluation provides deeper insight than end-to-end metrics.

**Questions:**

1. Were the six error types derived from any minded empirical study or developer survey of configuration failures?

2. Did you experiment with multiple variants per error type to check whether the evaluation metrics remain stable?

3. How does EnConda-Bench handle repositories with pre-existing errors or ambiguous READMEs?

---

> ### Author Response · Authors · 2025-11-22
> **Response to Reviewer qXby (Part 1/3)**
>
> We sincerely thank you for your insightful comments and constructive feedback, and we appreciate the time you took to review our paper. We hope our response below addresses your concerns.
>
> ## Response to Weaknesses:
>
> > The six error types chosen are said to be "guided by failure modes frequently encountered in practice", but no citation or prior empirical study is provided to substantiate this taxonomy. Without grounding in developer-observed data, it is unclear whether these six types cover real-world failure modes, or simply reflect intuitive assumptions.
>
> **Response**：
>
> Thank you for this suggestion. We agree that providing a basis for our taxonomy strengthens the benchmark design. In fact, our error taxonomy was derived through a rigorous, multi-step process:
>
> 1.  **Literature Review:** We surveyed existing empirical studies and review articles on configuration errors in software systems (e.g., [1]) to identify historically documented failure modes.
> 2.  **Community Mining:** We collected and synthesized common configuration issues reported by developers on StackOverflow and GitHub issues to ensure coverage of modern development challenges.
> 3.  **Expert Validation:** We conducted interviews with **5 expert engineers**, each with over 10 years of development and management experience. Based on their feedback, we removed non-typical error definitions and merged correlated types.
>
> Through this synthesis, we defined the six error types to cover the most prevalent real-world failure modes. Notably, we included an **"Other"** category to ensure that less common but possible errors are also represented. We have added this detailed derivation process to the revised manuscript (Page 13, Line 658-664).
>
> *[1] Yin Z, Ma X, Zheng J, et al. An empirical study on configuration errors in commercial and open source systems. SOSP 2011.*
>
> > Each erroneous README is created by injecting two errors per file. This raises two concerns: (i) since both synthesis and evaluation rely on LLM behavior, the resulting benchmark may reflect model-specific phrasing or error styles, rather than human-authored configuration errors; (ii) generating only two errors per category constrains diversity, a stronger design would sample multiple error candidates and retain a stratified subset verified by humans.
>
> **Response**：
>
> We apologize for the confusion regarding our generation and verification strategy. For diversity, our injection process does *not* hardcode "two errors per category." Instead, for each correct README, we generated 4-8 variant erroneous READMEs. We allowed the LLM to **autonomously select** error types from the taxonomy rather than enforcing a fixed distribution. After validation, we merged and split errors to create documents containing anywhere from **1 to 16 errors**. As shown in Figure 4, the final distribution of error types is relatively balanced, demonstrating that this autonomous strategy naturally covers diverse scenarios without artificial constraints.
>
> To address the concern that LLM-generated errors might reflect specific model biases rather than human errors, we conducted a test for errors. We randomly sampled 50 generated errors from our benchmark and collected 50 real-world configuration issues from GitHub. We mixed them and asked 3 senior engineers to classify each error as "AI-Generated" or "Real."
>
> **Table: Human Evaluation of Error Realism**
> | Actual Source | Judged as "Real" | Judged as "AI" |
> | :--- | :---: | :---: |
> | **Real-World Errors** | 58.0% | 42.0% |
> | **LLM-Generated (Ours)** | **54.7%** | 45.3% |
>
> The results show that engineers identified our generated errors as "Real" at a rate (54.7%) very similar to actual human errors (58.0%), suggesting our errors are indistinguishable from real-world ones. We have added this experimental result in the revised paper in Page 6, Line 308-312 and Page 7, Line 324-330.
>
> In addition to human verification, in our existing pipeline, we performed human verification on a sampled subset of the filtered data, achieving a **98.5% consistency rate**. While creating a fully human-verified stratified subset is resource-intensive, we acknowledge its value and plan to release a "Gold Standard" human-verified subset in future updates.

---

> ### Author Response · Authors · 2025-11-22
> **Response to Reviewer qXby (Part 2/3)**
>
> > The results lack ablation on error difficulty levels, number of injected errors, and impact of Docker environment variations. These would strengthen the claim that process-level evaluation provides deeper insight than end-to-end metrics.
>
> **Response**：
>
> Thank you for the suggestion. The ablation analysis of the number of injected errors is discussed in Figure 5. We found that the number of errors predicted by the model is **almost always more than the number in the ground truth**, and in some error types (such as E1), it can even be more than double. Further analysis revealed that the model almost **always detects 4-6 errors for a given erroneous README** during prediction, meaning it applies stricter criteria to uncover minor issues that have no real impact on the environment configuration. This is similar to the experience people have when using code agents in practice.
>
> Furthermore, We categorized the READMEs into 10 difficulty levels based on the number of injected errors. We observed an interesting phenomenon: model performance (F1-score) does not drop linearly with difficulty. Instead, it follows an **"inverted-U" shape**, peaking around Levels 5-7.
>
> **Table: Ablation Study on Error Difficulty Levels (F1-Score)**
> | Framework | LLM | **Level 1** | **Level 3** | **Level 5** | **Level 7** | **Level 10** |
> | :--- | :--- | :---: | :---: | :---: | :---: | :---: |
> | | | *P / R / F1* | *P / R / F1* | *P / R / F1* | *P / R / F1* | *P / R / F1* |
> | **Zero-shot** | GPT-4.1 | 27.6 / 40.2 / 32.7 | 38.7 / 46.4 / 42.2 | 74.2 / 44.6 / **55.7** | 84.2 / 35.0 / 49.4 | 88.4 / 20.6 / 33.4 |
> | | Claude-4 | 29.3 / 41.6 / 34.4 | 47.8 / 28.8 / 36.0 | 74.6 / 38.5 / **50.8** | 79.8 / 33.9 / 47.6 | 85.7 / 25.0 / 38.7 |
> | | Gemini-2.5 | 28.0 / 42.9 / 33.9 | 47.6 / 28.1 / 35.4 | 71.8 / 35.0 / 47.1 | 76.4 / 47.2 / **58.3** | 67.2 / 23.4 / 34.7 |
> | | DeepSeek-V3 | 27.5 / 34.3 / 30.5 | 43.0 / 20.5 / 27.8 | 64.7 / 32.3 / **43.1** | 71.0 / 29.6 / 41.8 | 71.4 / 26.3 / 38.4 |
>
> *(Note: We have condensed the table to representative levels for clarity.)*
>
> This trend correlates with our finding in Figure 5. Models tend to be **over-critical**, often predicting 4-6 errors regardless of the ground truth. Consequently, their predictions align best with the ground truth at Levels 5-7, resulting in the highest F1 scores. At lower levels (1-3), the high false-positive rate lowers the F1 score. We have added this in revised manuscript in Page 21, Line 1093-1109.
>
> For Docker Environment, we currently use a fixed base Docker environment to ensure reproducibility. We excluded dynamic Docker updates or varying base images because they introduce uncontrolled variables (e.g., network instability, deprecated dependencies) that make fair benchmarking difficult. We agree that handling dynamic environments is a real-world challenge and consider it a key direction for future work.
>
> ## Response to Question
>
> > Were the six error types derived from any minded empirical study or developer survey of configuration failures?
>
> Yes. Please refer to our detailed explanation in **Response to Weakness 1** above, where we outline the literature review, community mining, and expert interview process used to derive the taxonomy.
>
> > Did you experiment with multiple variants per error type to check whether the evaluation metrics remain stable?
>
> Yes. We did not restrict the generation to a single template. For each error type (Category), the LLM generated multiple specific variants (Sub-cases).
> For example, for **E6 (Logical Order Error)**, the generated variants include:
> - Instructing users to install dependencies (e.g., `pip install -r requirements.txt`) *before* activating the target virtual environment, causing packages to be installed in the wrong context.
> - Suggesting the installation of a high-level package *before* its required system-level dependency (e.g., running `pip install mysqlclient` before `apt-get install libmysqlclient-dev`), which leads to build failures.
> - Reversing the standard compilation sequence, such as attempting to run the test script (e.g., `python main.py`) *before* compiling necessary extensions (e.g., `python setup.py build`).
>
> To verify stability, we conducted an additional human evaluation. We sampled 10 specific errors per type and asked engineers to label them blindly. The human labels matched the generated tags **100%**, confirming that the metric and categorization remain stable across variants.

---

> ### Author Response · Authors · 2025-11-22
> **Response to Reviewer qXby (Part 3/3)**
>
> > How does EnConda-Bench handle repositories with pre-existing errors or ambiguous READMEs?
>
> Ensuring a clean set of original README was the first step of our pipeline:
> 1.  **High-Quality Filtering:** We selected repos with 10+ stars, 1000+ commits, and 10+ closed issues to ensure maturity.
> 2.  **Structure Check:** We filtered for repos with precise READMEs located in the root directory.
> 3.  **Human Verification:** Engineers manually attempted to set up the environment using the original README. **Any repository that failed to build or had ambiguous instructions was strictly removed.**
> This guarantees that every original correct README in our benchmark is correct and executable, and we have add this clarification to revised manuscript in Page 12, Line 630-642.
>
> ---
>
> All additional experiments and clarifications have been updated in the revised manuscript and are highlighted in blue, including clarification for error type definition (Page 13, Line 658-664), human evaluation of error realism (paper in Page 6, Line 308-312 and Page 7, Line 324-330), additional ablation studies (Page 21, Line 1093-1109) and clarification for repository selection (Page 12, Line 630-642). We once again thank you for your review and remain open to further discussion to improve our work.

---

> ### Author Response · Authors · 2025-11-28
> **Friendly Reminder for Discussion**
>
> As the rebuttal and discussion phases are drawing to an end, we wanted to gently follow up as we haven't received your feedback yet. If you have any questions or further comments regarding our responses, please feel free to share them. We are committed to addressing them during the remaining discussion period!

---

### Official Review · Reviewer_mVvg · 2025-11-01

**Soundness:** 3
**Presentation:** 3
**Contribution:** 3
**Rating:** 6
**Confidence:** 4

**Summary:**

This paper proposes a new dataset to advance agentic software engineering. This dataset focuses on measuring environment configuration, which is identified as a common weakness for current agents. It consists of 100 problem instances and includes process-level annotations. The dataset is constructed using LLM-assisted filtering as well as human validation.

**Strengths:**

* This paper identifies an important issue in agentic coding and proposes a targeted dataset and benchmark to enable future research.
* While there are no technical contributions beyond the dataset and some of the analysis, the evaluation suite does seem to offer some key benefits over previous work, in particular when it comes to the “process-level” evaluation.
* The dataset creation procedure is thorough and well-explained; I think this will be a useful resource for the community.

**Weaknesses:**

* There are some notable omissions in the evaluations, such a GPT-5-Codex and Claude 4.5, both of which are considered SOTA base models for coding. Furthermore, for the coding agents, why not include Codex CLI, Gemini CLI, Jules, and Claude Code? These are specifically optimized to handle novel codebases and deal with configuration issues.
* I’m not sure that ICLR is the best venue for this work. Perhaps a dedicated dataset/benchmark track would be better suited.
* Some of the figures are too small to be useful, such as Figure 5 and Figure 6. It would be better to focus on a subset of the results in the main text (relagate others to appendix) to better highlight differences.

**Questions:**

Why not include Codex CLI, Gemini CLI, Jules, and Claude Code? These are specifically optimized to handle novel codebases and deal with configuration.

---

> ### Author Response · Authors · 2025-11-22
> **# Response to Reviewer mVvg (Part 1/2)**
>
> We sincerely thank you for your insightful comments and constructive feedback, and we appreciate the time you took to review our paper. We hope our response below addresses your concerns.
>
> > There are some notable omissions in the evaluations, such a GPT-5-Codex and Claude 4.5, both of which are considered SOTA base models for coding. Furthermore, for the coding agents, why not include Codex CLI, Gemini CLI, Jules, and Claude Code? These are specifically optimized to handle novel codebases and deal with configuration issues.
>
> **Response**：
>
> Thank you for this valuable suggestion. In our original experimental setup, we aimed to consider the selection between **open-source models** (e.g., DeepSeek-V3) and **advanced closed-source commercial models** (e.g., Claude 4-sonnet, Gemini 2.5 Pro). Regarding agent frameworks, we prioritized the most **widely used open-source code agents** in the SWE domain (e.g., OpenHands, SWE-agent) alongside the **specialized environment configuration agent** to cover a broad spectrum. Due to the rapid evolution of the field and resource constraints at the time of submission, we missed some of the latest advanced models and commercial frameworks you mentioned. To provide a more comprehensive evaluation view, we have conducted additional experiments. While we could not cover every new models due to time limits, we selected the representative models and frameworks from your list.
>
> **New Experimental Results:**
>
> | Framework | LLM | **Error Type Judgment** | | | **Error Desc.** | **Fix Suggestion** | **End-to-End** |
> | :--- | :--- | :---: | :---: | :---: | :---: | :---: | :---: |
> | | | **Pre** | **Rec** | **F-1** | **ACC** | **ACC** | **Pass@1** |
> | **Base Model** | | | | | | | |
> | *Zero-shot* | GPT-4.1 | 33.4 | 90.6 | 48.8 | 39.6 | 18.2 | 1.5 |
> | | Claude-4-Sonnet | 37.1 | 80.6 | 50.8 | 45.1 | 28.5 | 3.1 |
> | | Gemini-2.5-Pro | 35.2 | 77.8 | 48.5 | 45.2 | 25.2 | 1.8 |
> | | DeepSeek-V3 | 33.2 | 65.8 | 44.2 | 39.7 | 22.3 | 3.3 |
> | | *GPT-5-Codex* | *40.5* | *88.2* | *55.5* | *49.8* | *32.4* | *6.3* |
> | **Code Agent** | | | | | | | |
> | *SWE-agent* | GPT-4.1 | 43.7 | 83.2 | 55.3 | 49.8 | 30.7 | 7.2 |
> | | Claude-4-Sonnet | 46.4 | 85.6 | 58.2 | 52.4 | 34.5 | 9.4 |
> | | Gemini-2.5-Pro | 45.1 | 92.5 | 56.8 | 50.2 | 32.2 | 7.8 |
> | | DeepSeek-V3 | 41.2 | 70.3 | 51.9 | 44.5 | 27.8 | 7.4 |
> | | *GPT-5-Codex* | *52.1* | *91.4* | *66.4* | *58.7* | *41.2* | *11.5* |
> | *Gemini-CLI* | Gemini2.5-pro | *48.5* | *94.2* | *64.0* | *54.6* | *38.9* | *13.2* |
>
> From the results, we observe that even with more advanced agents and models, significant challenges remain in environment configuration. The capability to diagnose and optimize configuration tasks to a level comparable to human engineers still requires further exploration.
>
> We have incorporated these new results and analysis into the revised manuscript (see Page 8, Line 378-402). We are committed to continuously improving the quality of our paper and will finalize these additions in the camera-ready version.
>
> > I’m not sure that ICLR is the best venue for this work. Perhaps a dedicated dataset/benchmark track would be better suited.
>
> **Response**：
>
> We appreciate your rigorous review and your commitment to maintaining the high standards of ICLR. We would like to clarify that our wrok was submitted to the **ICLR Dataset and Benchmark Track**, which we believe is the appropriate venue for this work. Our paper focuses on establishing a rigorous benchmark for evaluating agents in software engineering environments, which aligns with the track's goals. Furthermore, there have been such accepted work at ICLR. For instance, **EnvBench [1]**, which similarly focuses on automated environment setup tasks, was accepted to **ICLR 2025**. This indicates that our work is well within the scope of this conference.
>
> *[1] Aleksandra Eliseeva, Alexander Kovrigin, Ilia Kholkin, Egor Bogomolov, and Yaroslav Zharov. Envbench: A benchmark for automated environment setup. In ICLR 2025.*
>
> > Questions: Why not include Codex CLI, Gemini CLI, Jules, and Claude Code? These are specifically optimized to handle novel codebases and deal with configuration.
>
> Thank you for the suggestion! Please refer to our detailed response in **Response to Weakness 1 above**, where we explain the rationale behind our experimental setup and present the additional evaluation results for the suggested agents.

---

> ### Author Response · Authors · 2025-11-22
> **Response to Reviewer mVvg (Part 2/2)**
>
> > Some of the figures are too small to be useful, such as Figure 5 and Figure 6. It would be better to focus on a subset of the results in the main text (relagate others to appendix) to better highlight differences.
>
> **Response**：
>
> We sincerely apologize for the difficulty caused by the small figure sizes. Due to page limits, we initially attempted to compress as much information as possible into the main text instead of appendix.
>
> In the revised manuscript, we have **enlarged Figures 4 (Page 5, Line 251-269), 5 (Page 8,Line 410-421), 6 (Page 9, Line 432-445) snd 7 (Page 9, Line 466-480)** to ensure they are clearly readable. To accommodate these changes within the page limit, we have selected the most representative sub-figures for the main text and moved the complete set of figures to the Appendix (Page 20-21, Line 1043-1090). We are committed to further optimizing the layout to improve the reading experience.
>
> ---
>
> All additional experiments and clarifications have been updated in the revised manuscript and are highlighted in blue, including additional experiment results (see Page 8, Line 378-402) and better presentation of figures (Figures 4 (Page 5, Line 251-269), 5 (Page 8,Line 410-421), 6 (Page 9, Line 432-445) snd 7 (Page 9, Line 466-480), Appendix (Page 20-21, Line 1043-1090)). We once again thank you for your review and remain open to further discussion to improve our work.

---

> ### Author Response · Authors · 2025-11-28
> **Friendly Reminder for Discussion**
>
> As the rebuttal and discussion phases are drawing to an end, we wanted to gently follow up as we haven't received your feedback yet. If you have any questions or further comments regarding our responses, please feel free to share them. We are committed to addressing them during the remaining discussion period!

---

### Official Review · Reviewer_x88R · 2025-11-01

**Soundness:** 3
**Presentation:** 3
**Contribution:** 3
**Rating:** 6
**Confidence:** 3

**Summary:**

The paper presents EnConda-Bench, a new benchmark for evaluating llm code agents. The authors collected various repositories which includes neccessary README files which can be used as a guide to correctly setup the repository environments. The author synthetically introduce errors in the README using LLMs and create the benchmark to evaluate how llm code agents can repair the issues in the README as well as to generate a correct bash script to setup the environment. The paper evaluates several llm agent baselines and provides detailed analysis on the benchmark performance.

**Strengths:**

- tackles a very important and currently understudied problem of using llm agents to build software development environments
- the benchmark is constructed nicely with detailed descriptions of the pipeline and manual examination, which can be used by future work
- the authors also evaluate the benchmark already on several important baselines together with detailed analysis

**Weaknesses:**

Missing key evaluation category:
- As the author demonstrated in the paper and from prior work, developing a script that can successfully build an environment is non-trival.
- In the paper the authors focus on the task of repairing a README with errors
- However, we can also easily use the benchmark without README with errors to evaluate given a correct README what are the performance of generating a correct environment using LLM agents.
- I think this is an interesting scenario and can allow the authors to further compare and contrast the repair results

Unclearness of the error classification class:
- From reading the paper it is unclear how the error classification is done.
- For example how does an agent know what are the different error categories? if they do not know, then do the authors still use llm-as-judge to determine that?
- I think this is a very inefficient evaluation category as it only shows how good the LLM-agents are with identifying the error type

Minor issues:
- some of the figures are extremely small with small texts that are difficult to see (e.g., Figure 4)

**Questions:**

1. Did the authors evaluate the base error-free results from the benchmark?
2. How does the agent perform the error classification during evaluation?

---

> ### Author Response · Authors · 2025-11-22
> **Response to Reviewer x88R （Part 1/2)**
>
> We sincerely thank you for your insightful comments and constructive feedback, and we appreciate the time you took to review our paper. We hope our response can address your concerns.
>
> ---
>
> > Missing key evaluation category (Performance on correct READMEs)
> - As the author demonstrated in the paper and from prior work, developing a script that can successfully build an environment is non-trival.
> - In the paper the authors focus on the task of repairing a README with errors
> - However, we can also easily use the benchmark without README with errors to evaluate given a correct README what are the performance of generating a correct environment using LLM agents.
> - I think this is an interesting scenario and can allow the authors to further compare and contrast the repair results
>
> **Response**:
>
> Thank you for this valuable suggestion. As stated in the paper, our primary motivation for **injecting errors was to establish a "ground truth"** for environment configuration failures. Based on the specific error type, description, and the correct fix, we can perform a **fine-grained, process-level diagnosis of the agent's trajectories**. In contrast, when using a correct README, it is challenging to define specific failure causes or evaluate the complete execution trace with the same level of granularity.
>
> Additionally, we agree with your point that generating an environment from a correct README is an interesting scenario for comparison. We have conducted **additional experiments using correct READMEs for the selected models, keeping all other settings identical**. Since there are no error tags in correct READMEs, we evaluated the end-to-end success rate. The results are as follows:
>
> | Framework | LLM | **Error Type Judgment** | | | **Error Desc.** | **Fix Suggestion** | **End-to-End** | Correct README **End-to-End** |
> | :--- | :--- | :---: | :---: | :---: | :---: | :---: | :---: |:---: |
> | | | **Pre** | **Rec** | **F-1** | **ACC** | **ACC** | **Pass@1** |**Pass@1** |
> | **Base Model** | | | | | | | | |
> | *Zero-shot* | GPT-4.1 | 33.4 | 90.6 | 48.8 | 39.6 | 18.2 | 1.5 | 4.5 |
> | | DeepSeek-V3 | 33.2 | 65.8 | 44.2 | 39.7 | 22.3 | 3.3 | 6.3 |
> | **Code Agent** | | | | | | | | |
> | *SWE-agent* | GPT-4.1 | 43.7 | 83.2 | 55.3 | 49.8 | 30.7 | 7.2 | 9.2 |
> | | DeepSeek-V3 | 41.2 | 70.3 | 51.9 | 44.5 | 27.8 | 7.4 | 11.3 |
>
> We can observe that while the End-to-End Pass@1 rates on correct READMEs are slightly higher than those on the erroneous ones, the overall success rate remains surprisingly low. These results and analyses have been in the revised manuscript (Page 21, Line 1113-1133).
>
> > Unclearness of the error classification class:
> - From reading the paper it is unclear how the error classification is done.
> - For example how does an agent know what are the different error categories? if they do not know, then do the authors still use llm-as-judge to determine that?
> - I think this is a very inefficient evaluation category as it only shows how good the LLM-agents are with identifying the error type
>
> **Response**：
>
> We apologize for the confusion regarding the error type classification metric. We hope the following clarification helps explain how this metric works and its significance within our evaluation suite.
>
> We explicitly **provide the error type clarifications**, including their definitions and examples, in the agent's inference prompt (see details in Page 19, Line 979-990). Therefore, the agents are aware of the potential error types. Their task during execution is to detect these errors and classify them based on the provided definitions.
>
> Why this metric matters? The "Error Classification" metric **assesses the agent's perception capability**: Correctly identifying the error type is the foundation for generating an effective repair strategy.
>
> In addition, classification alone is insufficient. As shown in the first case of Figure 8 (Page 10), an agent might have only vaguely detected an error type but fail to pinpoint the specific issue. Therefore, we use "Error Description Accuracy" to evaluate whether the agent's specific diagnosis matches the ground truth as further assessment.
>
> In summary, Error Type Classification evaluates that whether the agent have the basic perception of the interaction environment, while subsequent metrics (like Description Accuracy and Fix Accuracy) evaluate the agent's precise feedback and action capabilities. Together, they form **a comprehensive process-level evaluation suite**.

---

> > ### Comment · Reviewer_x88R · 2025-11-22
> >
> > Thanks for the detailed response to my concerns and questions, including the additional experiments.
> >
> > I am curious about the new results. Why did the authors not evaluate any specific environment setup agents as they did in the paper?
> >
> > Also it seems that even the correct README results are still very low in terms of end-to-end success rate. This brings up the question of whether the agent/llm can even successfully setup the environment, let alone fix an incorrect environment readme file and then perform the correct setup. What are the author's insights regarding that?

---

> ### Author Response · Authors · 2025-11-22
> **Response to Reviewer x88R (Part 2/2)**
>
> > Minor issues: some of the figures are extremely small with small texts that are difficult to see (e.g., Figure 4)
>
> **Response**：
>
> We sincerely apologize for the difficulty caused by the small figure sizes. Due to page limits, we initially attempted to compress as much information as possible into the main text instead of appendix.
>
> In the revised manuscript, we have **enlarged Figures 4 (Page 5, Line 251-269), 5 (Page 8,Line 410-421), 6 (Page 9, Line 432-445) snd 7 (Page 9, Line 466-480)** to ensure they are clearly readable. To accommodate these changes within the page limit, we have selected the most representative sub-figures for the main text and moved the complete set of figures to the Appendix (Page 20-21, Line 1043-1090). We are committed to further optimizing the layout to improve the reading experience.
>
> ---
>
> All additional experiments and clarifications have been updated in the revised manuscript and are highlighted in blue, including additional experiments with correct READMEs (Page 21, Line 1113-1133) and better presentation of figures (Figures 4 (Page 5, Line 251-269), 5 (Page 8,Line 410-421), 6 (Page 9, Line 432-445) snd 7 (Page 9, Line 466-480), Appendix (Page 20-21, Line 1043-1090)). We once again thank you for your review and feel happy to further discussion to improve our work.

---

> ### Author Response · Authors · 2025-11-22
>
> We sincerely thank you for your prompt feedback and continued engagement with our work. We are glad that our previous response addressed your initial concerns. Regarding your new questions, we provide the following clarifications and insights:
>
> Regarding the evaluation of specialized agents (e.g., Repo2Run) on correct READMEs, due to the strict time constraints and computational resource limits during the rebuttal phase, we prioritized evaluating representative Base Models and General Code Agents to quickly establish a baseline trend for your reference. From the current results, we can already observe that the task remains highly challenging even for general agents. We are more than willing to conduct additional experiments with specialized agents to further validate these findings and will include the complete comparison in the final version of the paper.
>
> In addition to the low success rate, you raise a very thought-provoking point. The fact that end-to-end success rates are low even with correct READMEs highlights a critical reality: **Environment configuration is an unsolved, high-complexity problem in software engineering.** Unlike code generation, environment configuration involves complex interactions with the operating system, file systems, and external networks. Even for experienced human engineers, setting up a legacy project can take hours or days to solve version conflicts, OS-specific libraries, and non-deterministic factors (e.g., network timeouts or deprecated remote packages). The low success rate of current LLMs reflects this inherent difficulty. This "capability gap" actually underscores the necessity of our benchmark. If agents cannot reliably set up environments, they cannot effectively serve as autonomous developers. Our benchmark provides the granular diagnosis needed to understand *why* they fail (e.g., failing to detect a conflict vs. failing to execute the fix).
>
> Based on these insights, we believe future work can bridge this gap from two directions:
>
> 1.  **Agent Architecture:** Current agents often blindly execute commands. To improve, agents need richer capabilities similar to human engineers, such as **Enhanced Feedback Loops:** with better handling of execution errors to iteratively refine commands. Also, **External Knowledge Retrieval** can search the web or GitHub issues for solutions when encountering obscure build errors, rather than relying solely on internal knowledge.
> 2.  **Specialized Training:** Most LLMs are trained on code tasks, not environment configuration execution traces. Our framework can be used to generate large-scale, synthetic "configuration trajectories". This data can be used to train models specifically for the configuration domain, creating agents that "understand" the environment rather than just guessing commands.

---

> > ### Comment · Reviewer_x88R · 2025-11-22
> >
> > Thanks for the detailed feedback. I highly recommend the authors include the correct README setting in their benchmark, given the low score of correct README setting I think it is an important starting point for evaluation.
> >
> > I've increased my score to reflect this, thanks again.

---

> > > ### Author Response · Authors · 2025-11-23
> > >
> > > Thank you very much for recognizing our work! Your valuable comments have encouraged us to further refine our work. As you mentioned, we have incorporated the correct README settings into our new revised paper, see Page 8, Line 406-410 and Page 21, Line 1113-1133. Thank you again for your time and effort during the review process, and we would love to have any further discussion.

---

### Author Response · Authors · 2025-12-03
**Summary of Rebuttal and Discussion for Paper 6969**

Dear Area Chair,

We sincerely thank you for the time and effort dedicated to our submission. We provide this summary to help you access important information more conveniently.

**1. Core Contributions**

We believe our work fills important gap in the field of software engineering agent research because of the following contributions:

* In this paper, we propose EnConda-Bench, **the first environment configuration benchmark focusing on process-level trajectory evaluation**. Unlike existing benchmarks that only consider end-to-end success rates, EnConda-Bench precisely evaluates the agent's configuration trajectory. It diagnoses *where* the agent fails and *why* it cannot resolve the issue.

*   Environment setup has always required significant human effort and high-quality repository, making it a bottleneck for large-scale data synthesis. Our automated data synthesis pipeline is effectively helpful for **large-scale training trajectory data construction**.


**2. Summary of Reviewer Discussions & Revisions**

| Reviewer | Status | Key Concerns | Our Clarification, Supplement Experiments & Revised PDF |
| :--- | :--- | :--- | :--- |
| **R-x88R**    (initial 6) | **`Score Raised: 6 -> 8`**    (Concerns fully addressed) | 1. Missing baseline experiment on correct READMEs.    2. Clarity of error classification metrics.     | 1. Conducted **experiments on correct READMEs** (Page 21, Line 1113-1133), and provide insight about how to improve the agent in this challenge tasks in furthur discussion.    2.  Explained how **classification metrics assess agent** perception (Page 19, Line 979-990). |
| **R-mVvg**    (initial 6) | **Response Provided** | 1. Missing more advanced models (e.g., GPT-5-Codex).    2. Venue suitability (ICLR).     | 1. Added **model GPT-5-Codex and agent Gemini-CLI** (Page 8, Line 378-402).    2. Clarification and Cited related ICLR 2025 accepted work to prove scope alignment. |
| **R-qXby**    (initial 4) | **Response Provided** | 1. Error Type Definition Taxonomy.    2. Validity/Diversity of synthetic errors.    3. Lack of ablation studies. | 1. Details on how we determine the error definition, including the **literature review, community issue mining, and expert interview process** (Page 13, Line 658-664).    2. Added **human evaluation** shows synthetic errors are **indistinguishable** from real ones (54.7% vs 58.0%) (Page 6, Line 308-312 and Page 7, Line 324-330).    3. Added **ablation analysis** on error number and difficulty levels (Page 21, Line 1093-1109). |
| **R-pDzU**    (initial 6) | **Response Provided** | 1. Language coverage (Python only).    2. Realism of errors.    3. Concerns on Evaluation Metrics.    4. Confusion In Data Construction. | 1. Explaining the widespread use of Python and referencing related research.    2. Added **human evaluation** shows synthetic errors are **indistinguishable** from real ones (54.7% vs 58.0%) (paper in Page 6, Line 308-312 and Page 7, Line 324-330).    3. Clarification of metric of Error Classification, Fix Accuracy and Pass@1.    4. Provide more **details on filtering high-quality repositories**, including settings for human evaluation (Page 12, Line 630-642). |

**3. Conclusion & Acknowledgements**

During the discussion, we actively engaged with all reviewers. Notably, Reviewer x88R **raised the score from 6 to 8** after our response (23 Nov 2025, 01:32, before the information leak occurred). Although some reviewers did not participate in the discussion, we have **made every effort to clarify, supplement experiments, and revise the paper, and we believe this will resolve their concerns**.

Finally, thank you again for your dedication to maintaining the high standards of ICLR.

Sincerely,

The Authors of Submission 6969

---

### Meta-Review · Area_Chair_zAMM · 2026-01-06

**Summary:**

This submission proposes EnConda-Bench, an environment configuration benchmark for LLM code agents that emphasizes process-level trajectory evaluation beyond end-to-end success. Reviewers broadly agree the problem is important and underexplored, and they value the benchmark’s construction pipeline and the decomposition of evaluation into perception / feedback / action diagnostics. The main concerns raised across reviews were: (i) evaluation omissions (missing strong base models and specialized CLI agents; missing evaluation on correct READMEs), (ii) unclear or potentially limited evaluation design/interpretability (notably around error classification metrics and how error categories are used), (iii) validity/diversity/grounding of the synthetic error taxonomy and whether LLM-injected errors reflect real-world configuration failures, (iv) missing ablations (difficulty levels, number of injected errors), and (v) dataset construction transparency and presentation (repo filtering/verification details; figure readability).

After rebuttal, the authors added new baselines (including GPT-5-Codex and Gemini-CLI), added experiments on correct READMEs (end-to-end), provided grounding for the taxonomy (literature + community mining + expert interviews), added human studies on realism/label stability of injected errors, added ablations on error difficulty/number of errors, clarified repo filtering + manual verification, and improved figure readability. One reviewer (x88R) explicitly raised the score from 6 to 8 after these changes. Overall, the rebuttal substantively addresses most empirical/design transparency concerns and strengthens the case that this benchmark is a useful community resource for evaluating and diagnosing environment configuration failures.

**Reviewer Concerns:**

### Concerns addressed by the rebuttal / discussion

* **Missing “correct README” evaluation** (x88R): Addressed via added experiments reporting end-to-end performance on correct READMEs, and the reviewer explicitly recommended including this setting and increased their score after seeing it.
* **Omissions of advanced models / specialized agents** (mVvg): Addressed by adding GPT-5-Codex and Gemini-CLI results and incorporating them into the revised manuscript.
* **Unclear error classification metric / how agents know categories** (x88R): Addressed by clarifying that definitions/examples of error types are provided in the agent prompt, and positioning error classification as a perception metric complemented by error description and fix metrics.
* **Lack of grounding for the six error types / taxonomy legitimacy** (qXby): Addressed by adding the derivation process (literature review, community mining, expert interviews) and adding it to the manuscript.
* **Realism of LLM-injected errors / model-specific artifact risk** (qXby, pDzU): Addressed via a human evaluation comparing LLM-generated vs real GitHub errors (engineers labeled both as “real” at similar rates: 54.7% vs 58.0%).
* **Ablations on difficulty / number of errors** (qXby, pDzU): Addressed via an ablation over difficulty levels (based on injected error counts) and analysis of how performance varies with difficulty, plus discussion tied to model behavior (predicting 4–6 errors regardless of ground truth).
* **Handling repositories with pre-existing errors / README ambiguity** (qXby) and data construction transparency (pDzU): Addressed by clarifying repo maturity filters, root README filtering, and manual environment setup verification by 3 engineers with removal of repos that fail or are ambiguous.
* **Figure readability / presentation** (x88R, mVvg, pDzU): Addressed by enlarging key figures and moving full sets to the appendix.

### Concerns partially resolved

* **Breadth of agent coverage remains incomplete** (mVvg): While the authors added GPT-5-Codex and Gemini-CLI, the reviewer also suggested other specialized commercial agents (e.g., Codex CLI, Jules, Claude Code). The rebuttal expands coverage but does not claim comprehensive inclusion of all requested systems.
* **Interpretive value and cost of the process metrics** (x88R): The authors clarified how error classification is computed and why it matters, but x88R noted this metric may be “inefficient” and emphasized the importance of the correct README setting given low end-to-end success. The revised paper improves clarity, but the broader point that end-to-end setup remains very hard even without injected errors remains a limitation of current agents rather than the benchmark itself.
* **Docker/base-environment variation** (qXby): The authors explain they keep a fixed base Docker environment for reproducibility and defer varying base images/dynamic updates to future work; thus, robustness to environment variation is not evaluated in the current version.
* **Deeper analysis of specialized agent design choices** (pDzU): The authors offer a qualitative explanation (e.g., Repo2Run’s dual-environment monitoring and rollback/repair strategy) but do not add architectural ablations; they frame such analysis as future work.

**Reviewer Scores:**

* **Reviewer x88R (initial 6)**: The reviewer explicitly raised their score to 8 after the authors added the correct README setting and addressed the metric clarity and presentation issues.

* **Reviewer mVvg (initial 6,)**: The authors directly addressed the primary critique (missing strong models/agents) by adding GPT-5-Codex and Gemini-CLI results and clarified venue suitability as being the Dataset & Benchmark track. Figures were also improved. Since the reviewer did not provide follow-up, a conservative estimate is that they would maintain a positive score rather than materially increasing it, as the initial score was already above threshold.

* **Reviewer qXby (initial 4)**: The rebuttal directly targets qXby’s core concerns with: (i) explicit grounding for taxonomy (literature + community mining + expert interviews), (ii) clarifying generation diversity (4–8 variants per README; 1–16 errors after merge/split), (iii) human realism test vs GitHub errors, (iv) ablations on difficulty, and (v) clean-repo verification. Given these are the central reasons for the 4, it is plausible the reviewer would move to a borderline-accept stance.

* **Reviewer pDzU (initial 6)**: The authors address most concerns: realism of synthetic errors via human evaluation, partial credit via fix suggestion accuracy, repository filtering transparency with 3-engineer verification, and improved figures. However, limitations remain (Python-only scope; limited deeper analysis/ablations of specialized agents). The reviewer was already marginally above threshold; rebuttal likely reinforces that stance without necessarily warranting a large jump.

---

### Decision · Program_Chairs · 2026-01-26

Accept (Poster)